# Temporal-iCLIP captures co-transcriptional RNA-protein interactions

Ross A. Cordiner [1,2], Yuhui Dou [1,3], Rune Thomsen[1], Andrii Bugai [1], Sander Granneman [2] & Torben Heick Jensen [1] ✉

Dynamic RNA-protein interactions govern the co-transcriptional packaging of RNA polymerase II (RNAPII)-derived transcripts. Yet, our current understanding of this process in vivo primarily stems from steady state analysis. To remedy this, we here conduct temporal-iCLIP (tiCLIP), combining RNAPII transcriptional synchronisation with UV cross-linking of RNA-protein complexes at serial timepoints. We apply tiCLIP to the RNA export adaptor, ALYREF; a component of the Nuclear Exosome Targeting (NEXT) complex, RBM7; and the nuclear cap binding complex (CBC). Regardless of function, all tested factors interact with nascent RNA as it exits RNAPII. Moreover, we demonstrate that the two transesterification steps of pre-mRNA splicing temporally separate ALYREF and RBM7 binding to splicing intermediates, and that exon-exon junction density drives RNA 5'end binding of ALYREF. Finally, we identify underappreciated steps in snoRNA 3'end processing performed by RBM7. Altogether, our data provide a temporal view of RNA-protein interactions during the early phases of transcription.

The fate of an RNA polymerase II (RNAPII)-transcribed RNA is impacted by its dynamic association with RNA-binding proteins (RBPs). The early ribonucleoprotein (RNP) complex is acted upon by RNA processing-, transport- and decay-factors, which, dictated by transcript features, compete to shape RNP identity, while at the same time eliminate RNA processing by-products and mis- or excessively-produced material[1,2]. This process initiates during transcription where early remodelling steps impact RNP formation[3–5].

An early and omnipresent member of RNAPII-derived RNPs is the Cap Binding Complex (CBC) composed of CBP20 and CBP80. CBP20 binds nascent RNA after the transcript 5'end has received its hallmark 7-methylguanylate ($m^7G$) cap[6]. CBP80 in turn initiates the binding of proteins capable of directing the fate of the elongating RNA[2,7–9]. After productive transcription is initiated, the CBC is joined by the Arsenite resistance protein 2 (ARS2) to form the CBC-ARS2 (CBCA) complex[7,10,11], which then aids the recruitment of additional factors onto the nascent RNA (Fig. 1a). For multi-exonic transcripts, their

splicing was suggested to prepare the RNA for nuclear export[12] and the extent to which the CBCA complex vs. the splicing process contributes to attract the human Transcription and Export complex (hTREX) component ALYREF is being debated[13,14]. Adaptors of RNA decay factors are also loaded onto nascent transcripts to achieve RNA processing or complete degradation. One such adaptor is the trimeric Nuclear Exosome Targeting (NEXT) complex, which together with the ZC3H18 protein joins CBCA to form the CBC-NEXT (CBCN) complex[7]. Comprised of the MTR4, ZCCHC8 and RBM7 proteins, NEXT then recruits the ribonucleolytic RNA exosome to degrade a wide range of short non-adenylated transcripts and to process precursor forms of small RNAs, like snoRNAs, residing inside pre-mRNA introns[7,15–19].

While it is clear that the CBCA complex provides a basic platform for early factor recruitment, it remains elusive when and how these factors settle within their respective RNPs. Taking ALYREF as an example, it is unclear whether CBCA or the RNA splicing process serve to anchor ALYREF to the first exons of transcripts[13,20,21]. More generally,

[1]Department of Molecular Biology and Genetics, Aarhus University, Universitetsbyen 81, 8000 Aarhus, Denmark. [2]Centre for Engineering Biology, University of Edinburgh, Mayfield Road, Edinburgh EH9 3JD, UK. [3]Present address: Friedrich Miescher Institute for Biomedical Research, 4058 Basel, Switzerland. ✉e-mail: thj@mbg.au.dk

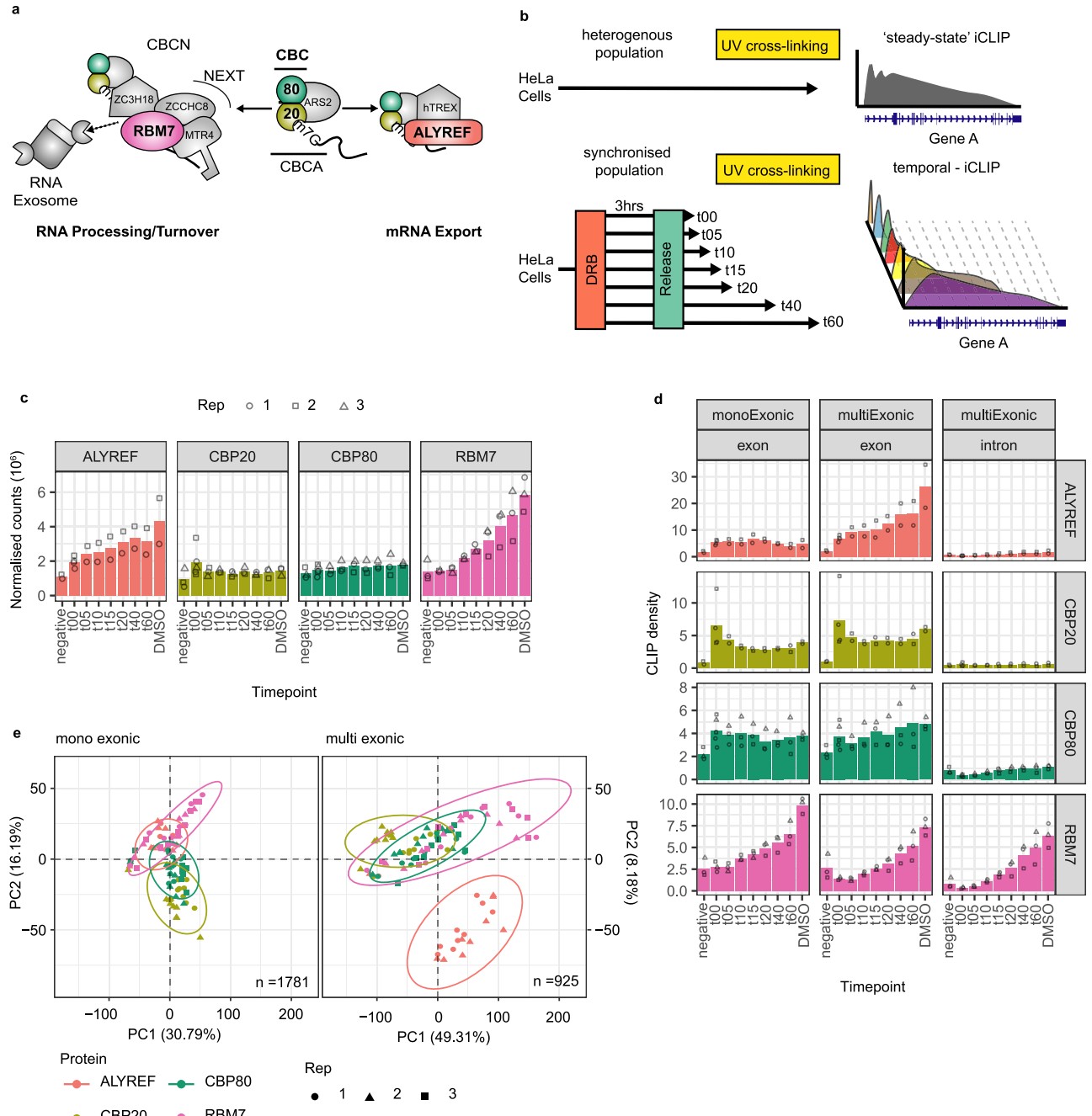

**Fig. 1 | Temporal-iCLIP uncovers dynamic RBP-RNA-binding profiles.**
**a** Simplified overview of the CBC and its two cofactors, ALYREF and RBM7, with relevance for the present study. See text for further detail. **b** Schematic outline of the tiCLIP approach (lower panel) as compared to regular steady state iCLIP (upper panel). **c** Histograms showing the average number of normalised mapped tiCLIP reads. Numbers from individual biological replicates are shown as circles, squares and triangles. 'Negative' timepoints represent controls in which blank magnetic beads were used (negative anti-GFP lanes) on unsynchronised samples.

**d** Histograms showing average RBP binding densities (reads/kb) calculated from all exonic and intronic regions of mono- and multi-exonic transcription units (TUs) as indicated. 22,650 TUs were used in this analysis. Biological replicates are shown as in **c**. **e** Principal component analysis (PCA) plot of tiCLIP data from mono (left)- and multi (right)-exonic TUs showing the variation across libraries. In order to capture the spatial variance, whole TUs were segmented into 10 kb bins and treated as individual data points. $n$ = number of TUs used for analyses. Biological replicates are shown as in **c**. Source data are provided as a Source Data file.

the process of splicing is complex, involving spliceosome remodelling and two transesterification steps, which first connects the 5′ splice site (5′SS) with the intron branchpoint (BP) and subsequently ligates the two exons while releasing the intron lariat. It is established that ALYREF settles upstream of exon-exon junctions[13,22], however, it is unknown exactly when during splicing ALYREF engages with nascent RNA. Finally, and conspicuously, ALYREF can also be found at the 3′ends of multi-exonic transcripts[22,23].

Like ALYREF, the NEXT complex is involved in both cap-proximal and -distal RNA transactions. In addition to its CBCN-mediated degradation of promoter-proximal transcripts, NEXT facilitates the exosomal processing of pre-snoRNAs[16]. Most often embedded within introns of host transcripts, these RNAs require splicing and lariat debranching for subsequent biogenesis. Here, NEXT facilitates the resection of 3′intronic pre-snoRNA sequences and is suggested to be recruited onto these cap-distal regions via spliceosomal

components sharing homology with RBM7 and ZCCHC8[24]. As for ALYREF, the exact stage of the splicing process upon when RBM7/ZCCHC8 anchor on the RNA is unknown.

As illustrated by the examples above, what molecular features cater to more stable RNP formation and when this occurs during transcription are understudied processes. Proteins binding to RNA via its modification, structure or specific sequence are well known, but many RBPs display promiscuous RNA-binding. Therefore, low-affinity RNA-binding might serve to locate higher affinity sites where local concentrations of proteins and RNA influence binding kinetics. Direct in vivo investigations of such dynamic RNA-protein interactions have so far been limited[25–27]. An established method of interrogation is UV cross-linking and immunoprecipitation (CLIP) coupled with high-throughput RNA sequencing[28]. UV irradiation covalently links RNA-RBP complexes at their sites of interaction, enabling recovery of spatial information of the given RBP throughout the transcriptome. A number of derivatives of the original protocol exist, which offer individual nucleotide resolution (iCLIP)[29], enhanced crosslinking efficiency[30] and reduced labour time[31–33]. Presently, however, these CLIP methods provide steady-state RNA-RBP binding information without temporal resolution. Instead, chromatin immunoprecipitation (ChIP) of RBPs and sequencing of associated nucleic acids have been used to describe co-transcriptional loading of RBPs throughout gene loci. Here, transcriptional synchronisation, using chemicals such as 5,6-Dichloro-1-beta-Ribo-furanosyl Benzimidazole (DRB), has been successfully employed to capture transcription kinetics in response to protein depletion or genetic insults[34–37]. Still, while ChIP localises a given RBP to chromatin, the technique is confounded by resolution limitations as the chemical cross-linkers employed do not provide direct RNA-RBP binding information.

Here, we combine transcriptional synchronisation with iCLIP to capture RNA-RBP binding profiles during early and steady-state phases of transcription. Using this temporal-iCLIP (tiCLIP) approach, we find that the CBC, ALYREF and RBM7 all associate across nascent RNA before establishing their distinct binding profiles. Moreover, our data suggest that specific metabolic events, such as the transesterification steps of splicing, discriminate RBP recruitment.

## Results

### tiCLIP reveals dynamic changes in early RNA-RBP binding

In order to capture dynamic RNA-RBP binding profiles, we developed tiCLIP, combining transient DRB-induced inhibition of RNAPII transcription elongation followed by time-resolved UV cross-linking of samples after the release of the DRB block (Fig. 1b, note comparison to 'steady state' iCLIP). UV cross-linking was applied from 5 to 60 min (t05-t60) post-DRB release. Baseline samples were formed by cross-linking cells without removing DRB (t00), or after incubation with the equivalent amount of DMSO to capture steady-state RNA-RBP binding. To interrogate the CBC and two of its functionally diverse interactors[7,38,39], we performed at least duplicate tiCLIP experiments on cell lines harbouring 'localisation and affinity purification' (LAP)-tagged[40] versions of ALYREF, CBP20 and RBM7 (Fig. 1a). Near-endogenous expression, correct subcellular localisation, and robust immunoprecipitation conditions for all of the engaged LAP-tagged proteins were confirmed (Supplementary Fig. 1a–d). We also captured the indirect tiCLIP profile of CBP80 due to its strong interaction with CBP20 (Supplementary Fig. 1c, d, mid panels). Inspection of auto-radiograms, visualising immunoprecipitated RNA-RBP adducts, revealed that for RBM7, ALYREF and less pronounced for CBP80, DRB-addition reduced RNA-binding (t00), which gradually recovered towards steady state levels after lifting the transcription elongation block (Supplementary Fig. 1e). CBP20's binding in close proximity to the RNA 5′cap supposedly prevented its RNA-mediated 5′-phosphate radiolabelling. Following normalisation of tiCLIP library sizes, based on rRNA read counts at each timepoint (see Methods), a general increase

in RBM7-, and ALYREF-RNA-binding matched the autoradiogram densities (Fig. 1c and Supplementary Data 1). To ensure that the indirect tiCLIP profile for CBP80 was not deriving from cross-linked CBP20 RNP, we explored RNA lengths cross-linked to these proteins. The median insert size for CBP20 vs. CBP80 libraries differed by only 1–8nt (Supplementary Fig. 1f), which was insufficient to retard CBP20 RNPs and cause co-migration with CBP80 (see Methods). In line with the association of the CBC with capped nascent RNA before DRB-mediated pausing of RNAPII, DRB-release had little effect on global CBP20/CBP80 tiCLIP signals. Finally, for all libraries, except CBP20, reads mapping to introns increased across the time course, demonstrating tiCLIP captures RBP binding to nascent elongating transcripts (Supplementary Fig. 1g).

To first focus on overall RNA-protein interaction patterns, RBP binding to transcripts deriving from mono- vs. multi-exonic transcription units (TUs) were analysed (Fig. 1d) and stratified into different TU biotypes (Supplementary Fig. 1h). We excluded snRNA- and replication-dependent histone (RDH) RNA-TUs from this analysis as their transcription is unaffected by CDK9 inhibitors, such as DRB[41]. Consistent with an association of the CBC with the RNA 5′cap protruding from stalled RNAPII[6,42], CBP20 binding was generally enriched inside exonic regions with limited dynamic change across the time course (Fig. 1d and Supplementary Fig. 1h). Curiously, and in contrast to CBP20, CBP80's intron binding increased across the time course (Supplementary Fig. 1g), implying that CBP80 may interact with the nascent transcript downstream of the 5′cap/first exon. ALYREF, instead displayed gradually increased exon-binding, which was only apparent for multi-exonic and protein-coding transcripts (Fig. 1d and Supplementary Fig. 1h). Additionally, we normalised ALYREF- to CBP20-binding, which highlighted ALYREF's preference for multi- over mono-exonic RNAs (Supplementary Fig. 1i). Corroborating these findings, steady state ALYREF (DMSO) binding over multi-exonic genes and RNA exon content correlated more positively than the binding densities of the other factors tested (Supplementary Fig. 1j, see also[14,22]), and principal component analysis (PCA) revealed that the tiCLIP coverage for ALYREF over multi-exonic TUs was unique (Fig. 1e, note changed position of ALYREF samples in the left vs. the right panel). Taken together, this indicates that ALYREF recruitment occurs during transcription and that at least some of its steady-state RNA interaction is splicing-mediated. Finally, RBM7 displayed an equal increase in intronic and exonic binding across the time course (Fig. 1d and Supplementary Fig. 1h), with a slight bias towards mono- vs. multi-exonic RNA (Fig. 1d and Supplementary Fig. 1i).

With the generated profiles, we conclude that tiCLIP effectively captures nascent RNA-RBP binding events with the capability of reflecting preferences, or lack thereof, of early vs. steady-state associations.

### The CBC, ALYREF and RBM7 sample elongating transcripts

To determine when the interrogated RBPs interact with elongating transcripts, we next leveraged the temporal dimension of tiCLIP by calculating the average RNA-binding density for each tested factor across all TUs, split into 1 kb bins from transcription start sites (TSSs) to transcript end sites (TESs). This identified two principles, which were readily visible for CBP80 and RBM7, but not for ALYREF and CBP20. Firstly, over the time course, waves of RNA-RBP associations were produced that progressively invaded the interrogated TUs (Fig. 2a). Secondly, these waves proceeded at largely the same rate for the two proteins and were independent of gene length, which suggested a common principle of RNA-binding. Indeed, the approximate RNA-RBP association kinetics were calculated to be ~3.5 kb/min, mirroring previously documented RNAPII transcription velocities (Fig. 2a, note black vertical lines indicating the predicted rate of RNAPII transcription[43,44]). These binding patterns were also visible in individual genes (Supplementary Fig. 2a, b). When focusing on promoter-proximal regions, CBP20 tiCLIP reads displayed a dense enrichment

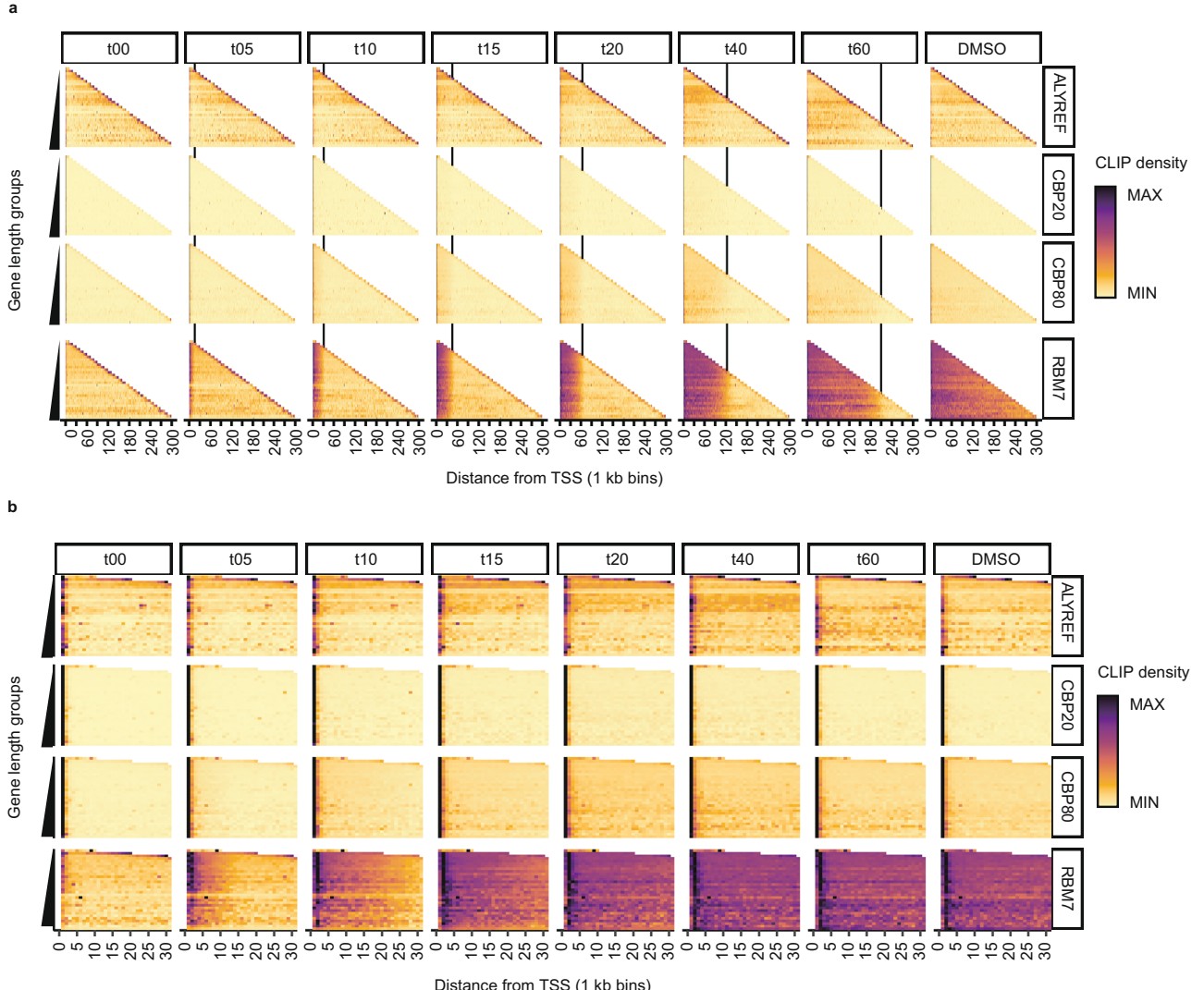

**Fig. 2 | Spatiotemporal RNA binding of the CBC, ALYREF and RBM7 is dictated by RNAPII transcription. a** Spatiotemporal heatmaps of all TUs generated using all tiCLIP reads. The values for each TU length group were normalised to the max value within the same gene length group. Each bin represents 1 kb. Black vertical lines mark the approximate speed of RNAPII transcription (3.5 kb/min). 23,512 TUs were used in this analysis. **b** Spatiotemporal heatmaps as in **a**, but zooming in on the first 30 kb of each TU. Source data are provided as a Source Data file.

restricted to transcript 5′ends (Fig. 2b, note first bin), whereas CBP80 binding was also detectable further downstream, supporting the idea that while CBP20 likely functions to anchor the CBC to the 5′cap, CBP80 may have broader access to RNA emerging from RNA-PII (Fig. 1d).

Lack of visible spatiotemporal RNA-RBP waves for CBP20 and ALYREF suggested that any nascent transcript binding might be masked by enrichment of CBP20 at RNA 5′ends and dominant exon binding preference of ALYREF, respectively. This is because reads mapping to exons may originate from stable RBP-RNA interactions preceding the DRB-block, and exonic regions make up a relatively small percentage of the TU; both of which prohibit the spatial resolution necessary to assess RNA-RBP binding. Hence, we generated spatiotemporal binding profiles using intronic reads only. Indeed, all interrogated proteins now bound RNA at similar rates (Supplementary Fig. 2c). The RNA-RBM7 interaction patterns were indistinguishable from the 'all reads' profiles (compare Fig. 2a and Supplementary Fig. 2c), corroborating the notion that initially, RBM7 has no binding preference for exons vs. introns (Fig. 1d). The intronic interaction profiles for ALYREF and CBP20 revealed their binding to nascent RNA (Supplementary Fig. 2c), and we, therefore, take this to suggest all

interrogated RBPs sample the nascent RNA co-transcriptionally before finding their longer residence binding sites.

## The CBC and ALYREF anchor relative to different transcript landmarks

Co-transcriptional sampling of nascent RNA is likely to be transient, while the high-density exonic binding, displayed by both ALYREF and the CBC, reflects their steady-state RNA associations. While it is presumed that CBC binding is restricted to 5′cap-proximal regions, ALYREF has been suggested to bind both at mRNA 5′- and 3′-ends as well as upstream of exon-exon junctions[22,23]. To interrogate this issue, we focused on exonic regions derived from multi-exonic TUs. As already demonstrated (Fig. 2b), CBP20- and to a lesser extent CBP80-RNA-binding was enriched in the first exons at steady state (Fig. 3a). In clear contrast, ALYREF displayed its primary steady-state binding within second exons and with decreased association with ascending exon number (Fig. 3b) and independent of the total exon count (Supplementary Fig. 3a). From iCLIP libraries, the exact site of an RNA-RBP interaction can be extrapolated from the 5′end of the read (the cross-linking site) (Fig. 3c, lower left), which for the bulk of both CBP20 and CBP80 binding was found to be within the TSS-proximal 50nt

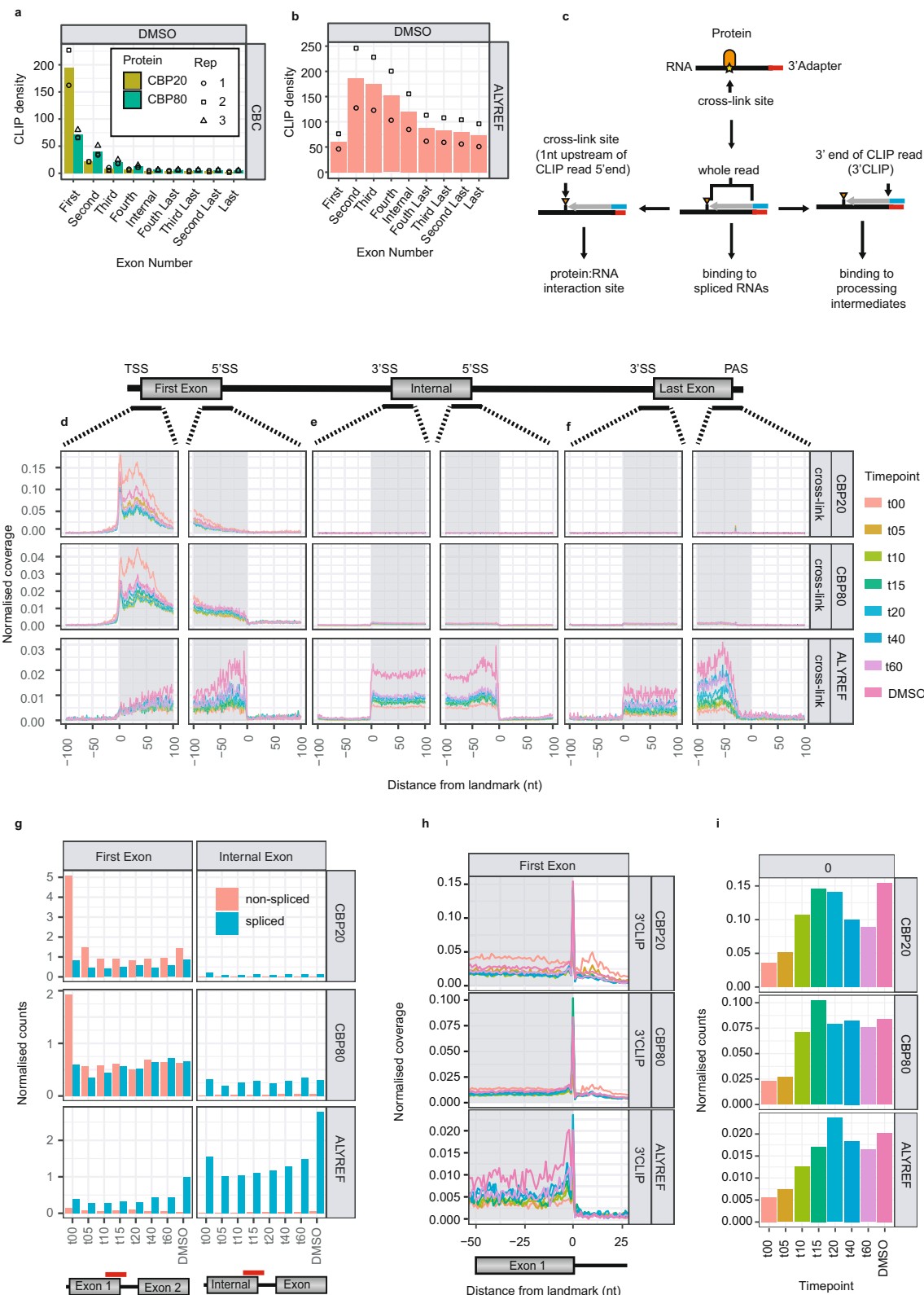

window (Fig. 3d–f and Supplementary Fig. 3b, upper and mid panels). These signals were enhanced by the DRB-block ('t00'), presumably due to promoter-proximal pausing of RNAPII. ALYREF binding was enriched at the 3'ends of exons (Fig. 3d–f, bottom panel), which was independent of the positional context of the exon or its size (Supplementary Fig. 3c). Moreover, whole read density profile enrichments could be observed in the 5'ends of internal exons (Supplementary

Fig. 3b, bottom panel), which was not visible in cross-link profiles (Fig. 3d–f, bottom panel). Taken together this is consistent with ALYREF binding to spliced RNA[22]. Interestingly, however, ALYREF cross-linking positions differed between those of first- and internal-compared to terminal-exons. For the former, peak ALYREF binding was detected at ~25 nt upstream of exon 3'ends, whereas it was at ~50 nt for the latter. This implies that pre-mRNA splicing and 3'end processing

**Fig. 3 | ALYREF and the CBC bind specific RNA splicing intermediates.**
**a**, **b** Histograms displaying the average 'steady state' CLIP (DMSO) binding densities for CBC (**a**) (CBP20 (yellow), CBP80 (green)) and ALYREF (**b**) across the first 4, the last 4 and any internal exons as indicated. Individual biological replicate samples are shown as circles, squares and triangles. 9998 TUs were used in this analysis. **c** Schematic representation of the CLIP read information displayed in coverage plots: cross-link sites (lower left), whole read (lower centre) or 3'end of CLIP read (3' CLIP) (lower right). Text indicates how the respective CLIP reads can be used to identify RBP binding to specific RNA species/intermediates. **d**–**f** Aggregate plots displaying the normalised coverage of cross-link sites across multi-exonic TUs for CBP20 (top), CBP80 (middle) or ALYREF (bottom), split into 201nt windows centred around the 5'- or 3'-ends of first (**d**), internal (**e**), or last exons (**f**). Shaded areas indicate exonic regions. Plots show the average signals from replicate samples. 11249 TUs were used in this analysis. **g** Normalised counts of non-spliced (red) vs. spliced (blue) reads, overlapping the last nucleotide within the first (left) or internal (right) exons. Below the histograms is a schematic representation of how non-spliced (red) vs. spliced (blue) reads, overlapping the ends of first or internal exons, were selected. Histograms display average counts from replicate samples. 3429 TUs were used in this analysis. **h** Aggregate plots displaying the normalised coverage of 3'CLIP data over a 76nt window centred around the 3'end of the 1st exon, for CBP20 (top), CBP80 (middle) and ALYREF (bottom). Shaded areas indicate exonic regions. 10752 TUs were used in this analysis. Timepoint colour code as in **d**–**f**. **i** Histograms quantifying the 3'CLIP signal present at the 3'ends of 1st exons ('0') in **h**. Average of biological replicates is shown. 4915 TUs were used in this analysis. Source data are provided as a Source Data file.

contribute differently to the positional recruitment of ALYREF, which is probably a reflection of exon junction complex (EJC)- and PABPN1-anchoring of ALYREF, respectively[22,23].

With the steady state binding patterns for ALYREF and the CBC on multi-exonic transcripts identified, we next addressed the kinetics by which these individual binding sites became occupied. Association of the CBC with nascent RNA was confirmed by a high ratio of non-spliced to spliced reads in the first exon at the transcriptional block, which was not seen during later timepoints or in internal exons (Fig. 3g; compare t00 vs t05-t60 for upper and middle panel). Commencement of splicing, after liberating cells from DRB, was the likely cause of this and implies that CBC binding does not require splicing and that its recruitment does not depend on active transcription. Conspicuously, a similar shift was not seen for ALYREF, suggesting that this protein is primarily recruited to spliced RNA (Fig. 3g, bottom panel). Due to the dynamic change in the abundance of non-spliced reads bound by the CBC within five minutes of activating RNAPII transcription, we reasoned that tiCLIP could capture the CBC bound to splicing intermediates resulting from first intron splicing. To pursue this, we exploited the fact that the pile-up of diverse 3'ends of CLIP reads (3' CLIP) may reflect different RNA processing intermediates bound by the RBP of interest (Fig. 3c, bottom right). Hence, we aggregated steady state 3'CLIP positions to generate a 76nt profile anchored over the 3'ends of first exons (Fig. 3h), and displayed the 3'CLIP data mapping to the last nucleotide of the first exon in histogram format for all tiCLIP timepoints (Fig. 3i). Indeed, CBP80 samples displayed 3'CLIP signals aligning with 1st exon 3'ends, indicative of RNAs having completed only the first transesterification reaction of splicing, leaving the exon 3'end unconnected to its downstream exon. Upon release of the DRB-block, this CBP80 3'CLIP density peaked at t15 before returning to steady state (DMSO) levels over subsequent timepoints (Fig. 3i, middle panel), indicating that many first introns were spliced during a short time frame after transcription reactivation. The 3'CLIP signal profile for CBP20 was qualitatively similar (Fig. 3i, upper panel).

Interestingly, the ALYREF-derived 3'CLIP profile displayed the same 1st exon hallmarks as for the CBC proteins (Fig. 3h, i, lower panel). However, it did not form a continuum of signal into the downstream exon, which would be expected of a protein preferentially binding to spliced RNA (Supplementary Fig. 3d). This demonstrated that ALYREF can be recruited before the second transesterification step of splicing, which was supported by the ability of the protein to co-precipitate components of the core-EJC (Supplementary Fig. 3e) recruited during the first transesterification step[45].

Our data, therefore, suggest, that while the CBC is bound before pre-mRNA splicing, ALYREF is anchored upstream of the exon-exon junction after the first transesterification step (Supplementary Fig. 3f).

### ALYREF anchoring on processed RNA is driven by two separate mechanisms
We next sought to characterise the mechanisms facilitating ALYREF anchoring and therefore plotted the distribution of its cross-linking

sites across exonic portions of all transcripts. This revealed enriched ALYREF binding across the cap-proximal ~25% and the 3'end 10% of transcript regions (Fig. 4a). Since such metagene representation might obscure specific sub-profiles harbouring unique ALYREF-RNA-binding, we performed a k-means clustering analysis of the profiles produced by all TUs, using each biological replicate of the ALYREF-DMSO samples independently. After intersecting these results (Supplementary Fig. 4a), two different RNA-binding profiles were immediately identifiable (Fig. 4b and Supplementary Data 2). The first ('group 1'), presented a strong 5'end enrichment of ALYREF with a slight additional peak at transcript 3'ends (Fig. 4b red signal and Supplementary Fig. 4b), whereas the second ('group 2') displayed only 3'end enrichment (Fig. 4b blue signal, and Supplementary Fig. 4c). Feature analyses uncovered that group 1 TUs displayed lower expression levels[46] (Fig. 4c, upper left panel) and were also generally longer than group 2 TUs (Fig. 4c, lower panels), but there was no difference in exon numbers between the two groups (Fig. 4c, upper right panel). Interestingly, however, group 1 transcripts harboured shorter 1st exons (Supplementary Fig. 4d), which correlated with higher exon-exon junction densities within their cap-proximal 25% (Fig. 4d), likely driving ALYREF-RNA anchoring at transcript 5'ends.

It was previously suggested that recruitment of ALYREF to transcript 5'- and 3'-ends was mediated by the CBC and PABPN1, respectively[23]. To test this idea, we leveraged previous ALYREF CLIP datasets[23], which were performed in conjunction with the depletion of CBP80 or PABPN1. Strikingly, CBP80 depletion had no effect on the 5' end binding of ALYREF to group 1 transcripts (Supplementary Fig. 4e). Conversely, PABPN1 depletion decreased ALYREF binding to the 3' ends of group 1 and 2 transcripts. Altogether, these data support protein-mediated recruitment of ALYREF to RNA 3'ends, while the process of splicing, and not the CBC, appears to drive ALYREF anchoring at RNA 5'ends, albeit only for a subset of transcripts.

### RBM7 anchors on splicing by-products post-transesterification
We previously characterised the steady state RNA-binding profile of RBM7 and identified a major enrichment of the protein at the 3'-ends of introns[16], which might be facilitated by a component of the SF3b complex[24]. To provide a temporal description of this RBM7-intron binding we generated aggregate plots of deconstructed tiCLIP reads (Fig. 3c) centred around 5'- and 3'-ends of multi-exonic TU exons (Fig. 5a–c). As DRB was removed, RBM7 binding switched from being exclusively exonic to become dually intronic/exonic (Fig. 5a, compare 't00-t15 data'), which presumably was triggered by increased intron abundance upon transcription reactivation, rather than by changes in RBM7 RNA-binding specificities. The latter, however, occurred at later timepoints where RBM7 read densities became specifically enriched at 3'ends of introns and upstream of polyadenylation sites (PASs); mirroring the steady state profile of the protein (Fig. 5a). Such RBM7 enrichment likely reflects its targeted recruitment to specific transcript regions and/or by-products, whereas enrichment over exonic regions likely reflects passive recruitment of the protein during transcription.

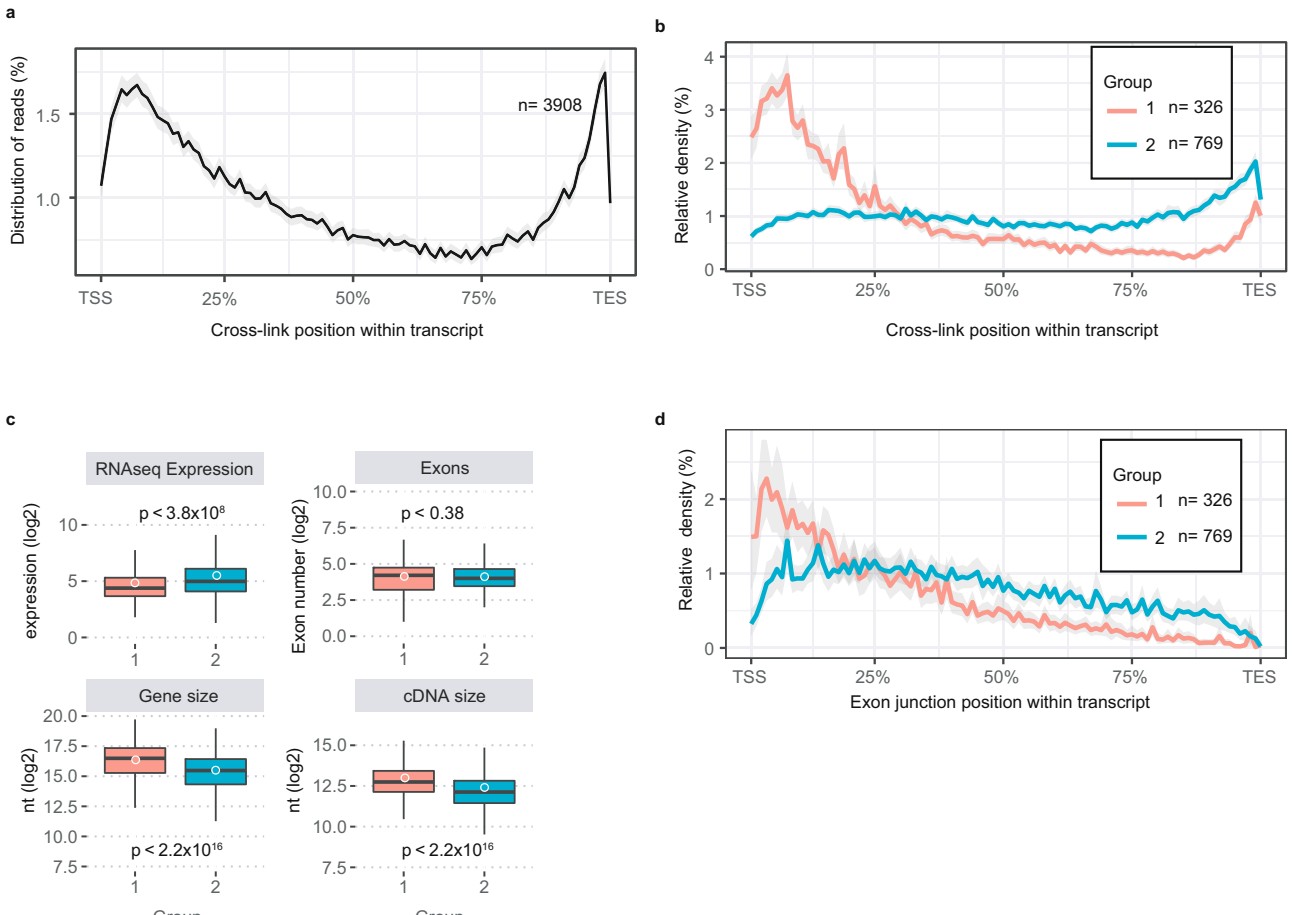

**Fig. 4 | Transcript features dictate different ALYREF anchoring profiles.**
**a** Average distribution plot of ALYREF cross-link sites across all TU's. TU's were normalised for length and expressing transcripts above 200nt. Data represent the average of two biological replicates, with confidence intervals shown as a grey ribbon. The number of TUs used for profiling is indicated. **b** As in **a** but stratified by gene groups identified by k-means clustering analysis (Supplementary Fig. 4a). **c** Boxplots indicating the median (middle line), first and third quartiles (box), ±1.5 × interquartile range (whiskers) and the sample mean (white circle) of RNAseq

expression values, total exons, gene size and cDNA size of the two gene groups from **b**. Shown values were log2 transformed. A two-sided Wilcox rank sum test was used to compare the means. No adjustments were made for multiple comparisons. *p* values are shown on respective panels. Group 1 *n* = 326 genes assessed over two biological replicates; Group 2 *n* = 769 genes assessed over 2 biological replicates. **d** As in **b** but plotting the average internal exon-junction densities across group 1 or 2 TUs. Source data are provided as a Source Data file.

Indeed, strong RBM7 cross-linking densities were identified in regions upstream (−100 to −20 nt) of the 3′ends of introns (Fig. 5b; note intronic region upstream of internal and last exon). These densities were followed by a decrease in RBM7 binding in a window immediately upstream of the 3′SS (−19 to −1 nt), which was much like that of the protein's exonic binding (Fig. 5b, note grey regions of 5′end of internal and last exon; positions 0 to 100nt). The shallow RBM7 binding in the −19 to −1 nt window was due to mapping restrictions of fragments below the 20nt size limit. Hence, cross-linking to this segment was only recovered in the absence of pre-mRNA splicing, and consequently, this highlighted that robust RBM7 intron association only occurs after the intron has been removed from its host transcript. This notion was corroborated by 3′CLIP data aligning at intronic 3′ends (Fig. 5c, note 5′ end of internal and last exon). Unexpectedly, we also identified an enrichment of RBM7 cross-linking sites at the 3′ends of the first and internal exons (Fig. 5b, note 3′end of first and internal exon). However, since iCLIP cross-linking sites denote 5′end positions of the produced cDNA with a 1nt upstream shift, this enrichment could derive from truncated cDNAs ending at intron 5′ends[29], which, given that RBM7 binds to the intron after splicing, seemed unlikely. An alternative explanation would be that RBM7 binds intron lariats. During iCLIP library preparation, the limited RNase I digestion of lariats can generate a three-way junction containing two 3′ends amenable for adapter

ligation and cDNA synthesis (Fig. 5d). Although both cDNA products will be truncated due to the BP, their respective 5′ends will map to the 5′end of the intron or to the BP, depending on whether they were reverse transcribed from inside the lariat or from the 3′end of the intron. Consequently, capturing these two signals is indicative of RBP-lariat binding[47].

This interpretation that RBM7 binds intron lariats was confirmed by cDNA truncation signals mediated by BP 2′−5′ linkages to 5′SSs (Fig. 5e, see signal at 'BP', and Supplementary Fig. 5a, b). Only at longer distances from downstream 3′SSs did cross-links appear; due to the aforementioned mapping constraints. Independent of the BP-3′SS distance, the 3′CLIP signals aligned with the last nt of the intron and increased in density over the time course (Fig. 5f); underscoring co-transcriptional recruitment of RBM7 to spliced introns. Although we confirmed that the signal associated with BP-mediated truncation was responsive to splicing, we note that exact truncation points were shifted 1-2nt upstream of expected BPs or 5′SSs (Supplementary Fig. 5c, d). As these truncations are caused by the same RNA structures and not polypeptide cross-links, we hypothesised that truncation patterns might differ depending on which part of the lariat the reads originated from. Hence, we focused on the mutational profile of 5′ends of reads that mapped to a 21nt window centred over BPs or 3′ends of internal or first exons. However, truncated reads originating from the

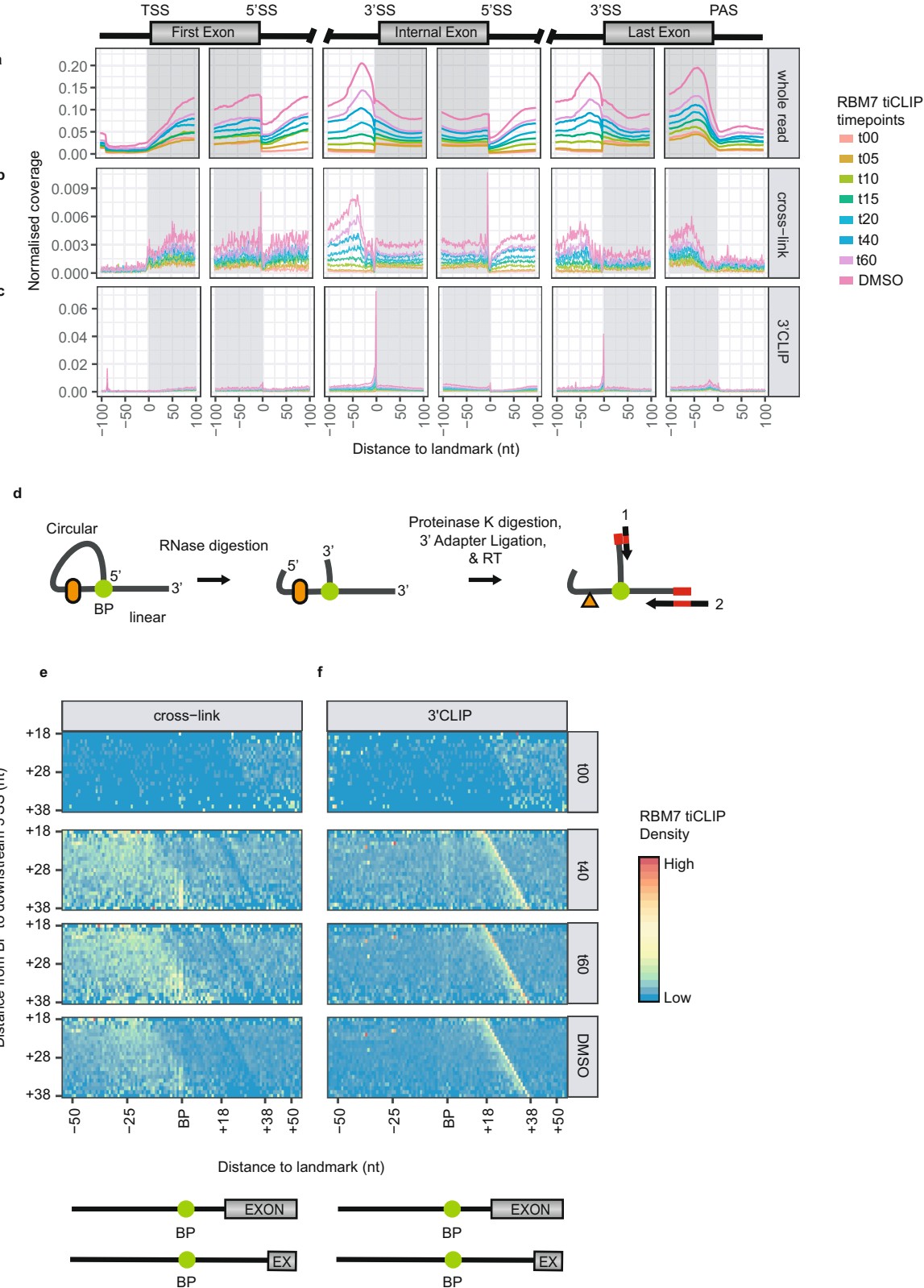

**Fig. 5 | RBM7 is recruited to introns before debranching but after the second transesterification step. a–c** Normalised RBM7 coverage profiles as in Fig. 3d, e, but across multi-exonic TUs and displaying the whole read (**a**), the cross-link site (**b**) or the 3′CLIP site (**c**). 11249 TUs were used in this analysis. **d** A schematic representation of the two possible cDNA truncation sites upon reverse transcription of a cleaved lariat intron. RT primer '1' indicates cDNA synthesis from within the lariat, whereas RT primer '2' indicates cDNA synthesis from the linear part of the lariat. Cross-linked protein is represented by orange cartouche, whereas the orange triangle represents a short peptide cross-linked to the RNA that remains after Proteinase K treatment. Green dots represent branchpoints (BP). **e, f** Heatmaps displaying RBM7 cross-linking (**e**) and 3′CLIP (**f**) sites centred around a 101nt window covering intron BPs, which were stratified by their ascending distance (18–38 nt) to the downstream 3′splice sites (3′SSs) and in vertical panels by their timepoints shown to the right. Schematic of BP to 3′SSs distance displayed below heatmaps. Timepoint or sample shown to right of heatmap. 20811 BPs were analysed. Source data are provided as a Source Data file.

circularised region of the lariat were more likely to truncate on the BP and be revealed by the high density of mismatches 1nt upstream of the intron (Supplementary Fig. 5e). Conversely, reads originating from the linear part were truncated before the BP, as revealed by the high density of correctly mapped 5′ends terminating 1nt downstream of the BP (Supplementary Fig. 5f). These differences were likely due to the incompatible 2′−5′ phosphodiester linkage used as a template for cDNA synthesis, allowing incorporation of a terminal nucleotide when the read originates from the circular region (Supplementary Fig. 5g). Instead, the 2′−5′ phosphodiester linkage completely blocks reverse transcription when the read originates from the linear region (Supplementary Fig. 5h), which would explain the shift in truncation patterns observed in Supplementary Fig. 5c, d.

To summarise, we were able to capture early, mid and late RBM7-RNA-binding profiles. RBM7 was found to bind transcripts immediately following their exit from RNAPII, and later became focused on binding to the 3′end regions of introns. Here, anchoring occurred after the second transesterification step of splicing, but before lariat debranching.

### RBM7 tiCLIP reveals snoRNA processing intermediates

Embedded within sequences of many mammalian RNAPII transcripts, snoRNAs are released from their host introns by splicing and matured by exoribonucleolytic resection of flanking intronic sequences[48]. Two classes of snoRNAs constitute different functional enzymes and are categorised by H/ACA or CD-box sequence motifs harboured within their mature sequences. RBM7, as part of NEXT, is essential for licensing decay of the 3′flanking intronic sequences of precursor snoRNAs (pre-snoRNA)[16]. To obtain a temporal view of this, we quantified RBM7 tiCLIP read coverage over 101nt windows centred around the 3′ends of annotated snoRNAs and their downstream host intron 3′SSs. While the steady state RNA-binding profile of RBM7 displayed a clear enrichment in the 3′intronic flank, this was abrogated upon the 3 hr DRB-block (Fig. 6a), which was likely due to snoRNA metabolism reaching completion during this time frame. Consistently, after DRB-release, RBM7 association with newly transcribed snoRNA-containing transcripts, progressively increased over 3′flanking intronic sequences (Fig. 6b). RBM7 binding to the 3′end of snoRNA-containing introns was similar irrespective of snoRNA subclasses. However, the binding profile differed between H/ACA and CD-box snoRNAs in a 25nt window extending from the 3′end of the snoRNA into the intronic RNA, suggesting RBM7 was bound to snoRNAs harbouring different lengths of 3′ extensions (Fig. 6a, b; see left panel). To investigate this further, we utilised 3′CLIP read information, which revealed RBM7 binding to precursor snoRNAs with uniform extensions (Fig. 6c, d; see various coloured arrows). For H/ACA snoRNAs, two 3′CLIP peaks downstream of the 3′end of the snoRNA were readily apparent and indicated that RBM7 was bound to 9nt and 25nt 3′extended H/ACA snoRNAs. For CD-box snoRNAs, a continuous high-density of 3′CLIP reads mapped from the 3′end of the snoRNA onto 25nt downstream, indicating that RBM7 was bound to CD-box snoRNAs with 3′extensions of variable length, but at a maximum of 25nt. These RBM7-derived 3′CLIP profiles were representative of multiple individual CD- and H/ACA-box snoRNAs (Fig. 6e, f and Supplementary Fig. 6a). Interestingly, the 3′CLIP peaks were also seen at intron 3′ends (Fig. 3c, d see right panel). Capturing these processing intermediates was likely the result of higher occupancy times on RNA, which highlighted where RBM7 was loaded to begin processing, or unloaded upon its completion.

The last 9nt of the 3′flanks of H/ACA snoRNA intermediates were recently shown to be processed in a NEXT-independent manner by EXOSC10[49]. An accumulation of 9nt extended snoRNAs bound by RBM7 might therefore uncover the limits of RBM7-mediated H/ACA snoRNA processing. In support of this, re-analysis of RNA 3′end-seq data, capturing both polyadenylated (pA+) and unadenylated (pA−) transcripts from HeLa cells depleted of the core exosome component RRP40[18] (see Methods), revealed an increased abundance of 3′ends immediately downstream of the 9nt and 25nt peaks identified by RBM7 3′CLIP analysis of H/ACA- and CD-box snoRNAs, respectively (Supplementary Fig. 6b). As similar increases were not present upstream of these peaks, we suggest they represent blocks to further exosome/RBM7-mediated processing (Fig. 6g).

Similar to generic introns, snoRNA-containing introns were also bound by RBM7 after the second transesterification step of splicing, as evidenced by high RBM7 3′CLIP density aligning to the 3′SSs (Fig. 6c, d; see right panel). However, the lariat binding hallmark of cross-link signal at the 5′end of the intron was absent in snoRNA-containing introns, possibly reflecting different RBM7 binding kinetics for snoRNA-containing vs. generic introns (Supplementary Fig. 6c). Furthermore, RBM7 binding to the intronic sequence upstream of the snoRNA did not increase over the time course, indicating that it is not involved in 5′end processing of pre-snoRNAs. Finally, we noted that RBM7 cross-linked to mature snoRNA sequences and that this increased over the duration of the time course (Supplementary Fig. 6d, e; see left panel), implying that RBM7 ends up bound to the mature snoRNA sequence during pre-snoRNA processing.

Taken together, these data illuminate the biphasic nature of snoRNA 3′end processing which is dictated by the occupancy time of RBM7 on RNA. Two regions on snoRNA loci where this occurs are at the intron 3′end and the region downstream of mature snoRNA. Higher occupancy times at the intron 3′end could be due to initial loading and slow decay by the exosome; whereas steric inhibition of exosome processing caused by the bulky snoRNP may cause higher RBM7 occupancy times downstream of the mature snoRNA.

## Discussion

We have employed tiCLIP, to capture the spatiotemporal binding of the CBC, ALYREF and RBM7 to RNA from the onset of transcription. Despite ALYREF and RBM7 informing opposing RNA fate decisions of nuclear export and processing/decay, respectively, a commonality between the tested RBPs is that their initial transcript binding correlates with the predicted speed of RNAPII transcription (Fig. 2a and Supplementary Fig. 2c). We suggest such early RNA-RBP association represents an initial phase of transient RBP searching for high affinity anchoring sites; a search process which can be visualised by the temporal resolution offered by tiCLIP. Illustrated by the proteins interrogated here, RBP anchoring may then be aided by a specific RNA motif/modification, a particular RNA processing event or via recruitment by other proteins. The tiCLIP methodology enables parallel processing of multiple experimental timepoints. Since this approach involves pooling of barcoded CLIP samples before their cDNA synthesis and library amplification, it allows the direct quantitative assessment of RBP IPs from varying conditions with significantly reduced resources and labour time.

tiCLIP captured the dynamic RNA-binding profile of CBP20 and, via its co-precipitation, CBP80. Through its interaction with the m7G cap, CBP20 anchors the CBC to the 5′end of all RNAPII transcripts[50–52], which was reflected by a strong cap-proximal CBP20 CLIP density (Figs. 2b and 3a). Interestingly, CBP80 CLIP signals extended beyond this 5′proximal boundary, indicating that the protein can interact with RNA distal to the m7G cap. Indeed, prior ChIP experiments demonstrated CBC binding within gene bodies[53–56]. Moreover, an interaction between CBP80 and the RNAPII C-terminal domain (CTD), phosphorylated at its Ser2 and Ser5 residues, was identified[57]. It is therefore possible that the CBC 'hitches a ride' with RNAPII to remain in close proximity to the exiting nascent RNA. In this configuration, CBP80 might be flexible to engage with downstream regions of nascent RNA to facilitate deposition of its cofactors, while remaining anchored to the 5′end of the transcript via CBP20.

ALYREF has been suggested to be anchored at the 5′ends of transcripts in a process mediated by the CBC[13,23]. However, our tiCLIP

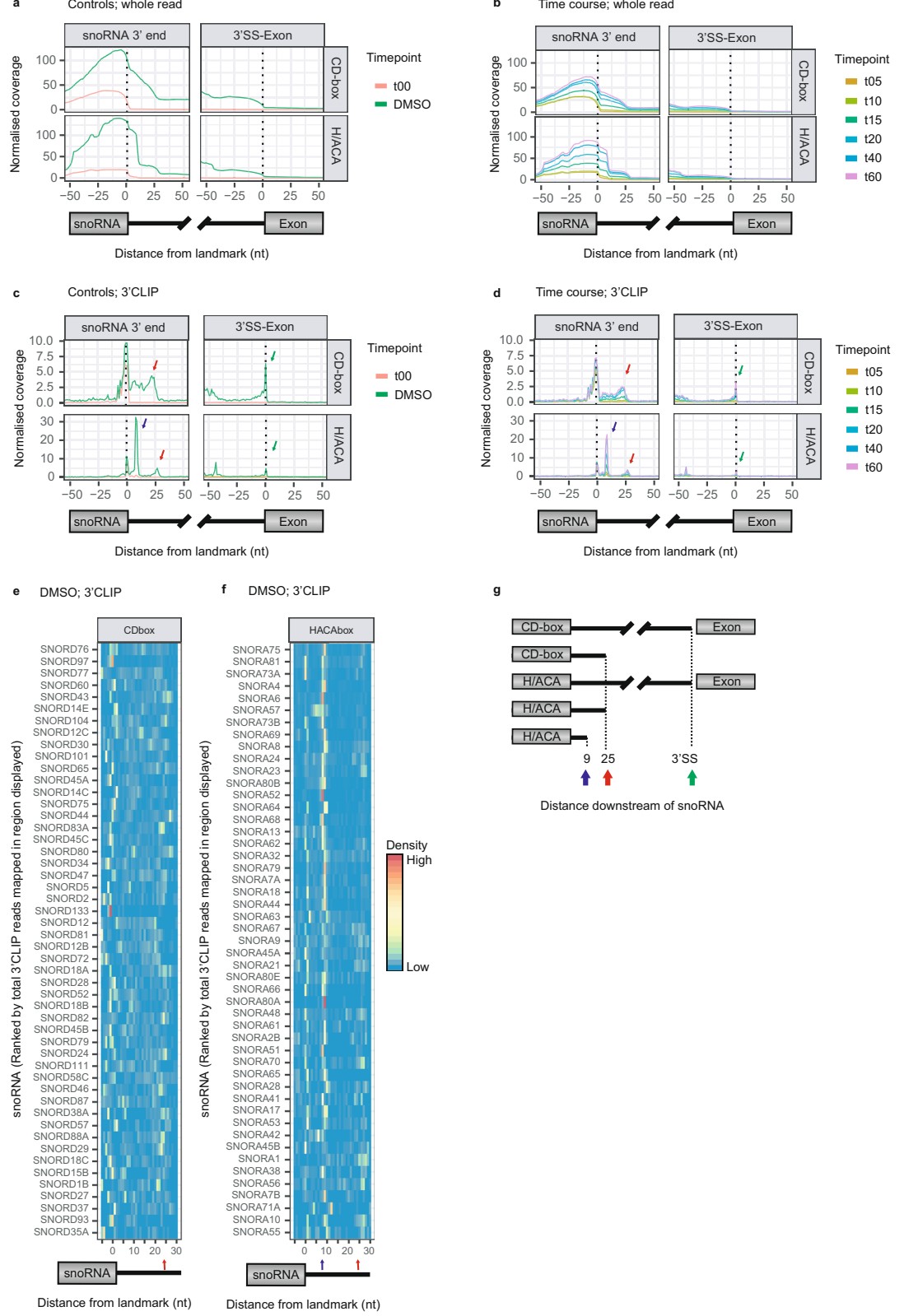

**Fig. 6 | RBM7 binds specific snoRNA intermediates. a, b** Whole tiCLIP read coverage of RBM7 plotted around a 101nt window centred on snoRNA 3′ends (left panel) or their downstream 3′SS (right panel). Profiles were stratified by CD- and H/ACA-box snoRNA classes as indicated. DMSO and t00 samples are shown in **a**, while all timepoints are shown in **b**. **c, d** as in **a, b** but plotting 3′CLIP data. Coloured arrows indicate common 3′CLIP data peaks. Blue colour is unique for H/ACA snoRNAs at 9nt. **a–d** 487 snoRNAs were used in this analysis. **e, f** Heatmaps depicting 3′CLIP signals downstream of CD-box (**e**) or H/ACA-box (**f**) snoRNAs. The 50 snoRNAs with the most cumulative total 3′CLIP signal present in the displayed 35nt window are shown. Coloured arrows as in **c, d**. **g** Schematic representation of 3′extended snoRNAs bound by RBM7. Arrows denote 3′extended snoRNAs as also shown in **c–f**. Source data are provided as Source Data file.

analysis contrasted this idea and leads us to conclude that the pre-mRNA splicing process is the main driver of ALYREF anchoring; also, at transcript 5′ends. This is based on three lines of evidence. First, low ALYREF coverage was detected over mono-exonic RNAs (Fig. 1d and Supplementary Fig. 1h), which was supported by re-analysis of published ALYREF CLIP datasets[22,58]. Second, the density of promoter-proximal exon-exon junctions correlated strongly with the 5′end enrichment of ALYREF (Fig. 4b, d). Third, depletion of CBP80 did not impact 5′enrichment of ALYREF on transcripts with 5′proximal exon-exon junctions (Supplementary Fig. 4e)[23]. It was recently suggested that ALYREF uses the CBC as an intermediate loading point before its anchoring upstream of the EJC, which was based on the finding that a seemingly transient CBC-ALYREF interaction was stabilised by blocking transcription[22]. However, a direct impact on ALYREF-RNA-binding was not assessed. In our experiments, we failed to detect cap-proximal ALYREF-RNA-binding during the DRB-mediated transcriptional block (Fig. 3d; lower left panel). If the CBC indeed provides for an intermediate loading of ALYREF, this presumably occurs then solely by protein-protein interaction. More generally we suggest that ALYREF represents an important landmark to define spliced RNA in preparation for its nuclear export. During early phases of transcription, premature termination is a common outcome leading to the production of abundant amounts of mono-exonic RNA[46,59–61]. As much of this material needs to be removed by the nuclear RNA exosome[2,5], it would be desirable to avoid binding by a nuclear export factor. However, upon intron definition and assembly of the spliceosome, transcription elongation becomes more stable, and nascent RNA now acquires ALYREF and possibly other proteins necessary to form export-competent RNPs.

The temporal resolution of tiCLIP enabled us to establish that ALYREF anchoring occurs upstream of the exon-intron boundary after the first transesterification step of splicing before the full completion of the process. This conclusion was drawn from three main lines of evidence. First, CBC 3′CLIP reads mapped to transcript first introns during the DRB-mediated transcriptional block and throughout early time points after its release, whereas the cognate ALYREF 3′CLIP read profile was exon restricted (Fig. 3h; compare upper and lower panels unshaded areas). We take this to indicate that the CBC binds prior to the commencement of splicing, whereas ALYREF enters after its initiation. Second, both ALYREF and CBC 3′CLIP reads mapped to the 3′ ends of 1st exons, suggesting that both proteins are bound to 1st exons that have undergone the first transesterification step of splicing (Fig. 3h; see '0nt'). Thirdly, 3′CLIP densities for CBP80 and ALYREF were higher during the transcriptional time course than at steady state, indicating a synchronised wave of simultaneous TSS-proximal splicing events after restarting transcription. These dynamic data corroborate that the CBC anchors to the nascent transcript early, whereas ALYREF binds splicing-dependently (Fig. 3i). Gratifyingly, the first transesterification step of splicing also coincides with the recruitment of the EJC components MAGHO, Y14 and eIF4A3[45] (Supplementary Fig. 3e), which end up being anchored 24nt upstream of the resulting exon-exon junction[62] and have been shown to interact robustly with ALYREF[63]. It, therefore, appears that preparation of the mRNP for nuclear export is launched already during the initial step of pre-mRNA splicing.

Our tiCLIP data demonstrated that RBM7 had a slight preference for mono- over multi-exonic transcripts (Fig. 1d and Supplementary Fig. 1i). For multi-exonic transcripts, RBM7 initially binds with no preference for intronic vs. exonic sequence (Figs. 1d and 5a). As previously suggested[16], such low-specificity RNA-binding by RBM7 probably operates as a fail-safe mechanism for transcript removal in case of the appearance of an unprotected RNA 3′-OH; a prerequisite for the RNA exosome to engage a substrate. As many NEXT substrates are products of ubiquitous promoter-proximal transcription termination events[18,46], we suggest that promiscuous RNA-binding by RBM7 bypasses the evolutionary need for specific motifs to license RNA decay hereby

providing the needed flexibility. In noticeable addition to its general RNA-binding, RBM7 extensively anchors to splicing intermediates, which tiCLIP identified to be introns having undergone the second transesterification step of splicing but many of which were not debranched. This conclusion was derived from two distinctive RBM7 tiCLIP patterns around exon-intron and intron-exon boundaries. Firstly, RBM7 3′CLIP reads primarily mapped to the 3′termini of introns (Fig. 5c, see internal exon), indicative of their post-spliced nature. Secondly, cDNA truncations, representing the 5′ends of introns being linked via 2′−5′ phosphodiester bonds to intron BPs, were detected (Fig. 5b, e; see 5′SS and BP, respectively). Anchoring of RBM7 is likely coordinated with the splicing machinery given its proximity to the 3′ end of the intron and to the BP. Indeed, RBM7 was previously reported to interact with SAP145/SF3B2[15,24], a component of the SF3b complex critical for recognising the BP. A proline-rich segment in SAP145/SF3B2 mediates its mutually exclusive binding to RBM7 or the SF3b component SAP49/SF3B4. Hence, RBM7 might replace SAP49/SF3B4 after its role in the first transesterification step of splicing. Being positioned prior to lariat debranching, it is tempting to speculate that RBM7/NEXT might be involved in this activity. However, mass spectrometry analysis of proteins co-purifying with RBM7 did not detect the debranching enzyme DBR[15]. Additionally, if debranching was mediated by RBM7/NEXT, depletion of NEXT components would predictably result in the accumulation of 5′extended snoRNAs, which was not supported by ZCCHC8 knockdown/RNAseq data[16]. We, therefore, suggest that RBM7 is mainly recruited to spliced-out introns to facilitate intron degradation/trimming.

A prime reason for RBM7 remaining on spliced-out introns appears to be its critical role in 3′end processing of intron-residing snoRNAs. To this end, our tiCLIP data uncovered hitherto undisclosed phases of snoRNA processing: RBM7 bound H/ACA-box pre-snoRNAs, that were 3′end extended by 9 or 25 nt, and CD-box pre-snoRNAs with variably sized 3′end extensions of maximally 25 nt (Fig. 6c, d). What might these 3′extended forms tell us about the role of NEXT in snoRNA processing? UV crosslinking captures direct RNA-RBP interactions with a slight preference for U-rich RNA sequences[64]. Since the highest density of RBM7 footprints was detected at intron 3′ends, which are inherently U-rich, longer RBM7 dwell times might not be the only reason for such strong cross-linking[65]. However, since sequences immediately downstream of mature snoRNAs are likely to be less U-rich, their high-density RBM7 binding likely reflects longer RNA dwell times of the protein, which could represent sequential RBM7-mediated loading of the RNA exosome onto the pre-snoRNA target. Supporting this idea, the RBM7-immunoprecipitated RNA species are reminiscent of those co-precipitating with a catalytically dead version of EXOSC10, which was suggested to capture substrates arising from handover events from processive exosome core-dependent DIS3 degradation to EXOSC10-mediated distributive processing[66,67]. This distributive RNase activity of EXOSC10 might require multiple exosome-loading events mediated by NEXT. In a non-mutually exclusive alternative, high-density RBM7 binding to pre-snoRNA 3′regions might result from continuous NEXT disassembly and reassembly on the RNA template. In this scenario RBM7 would be 'pushed' in front of an active exosome complex, stalling at sequences more difficult to degrade or where RBM7/NEXT progress is hindered by bulky RNP or structured RNA such as those manifested by the upstream snoRNP. Specific cross-links would then identify the dwell positions of RBM7/NEXT, and 3′CLIP profiles would reflect the RNA 3′ends residing within the NEXT-exosome holoenzyme. In support of this possibility, snoRNP components presumably bind their intronic snoRNA targets co-transcriptionally hereby being capable of blocking exosome progress[68]. As the tiCLIP data imply, the terminal position of NEXT, during snoRNA processing, would then be inside the mature snoRNA sequence (Supplementary Fig. 6d, e), consistent with the 3′end snoRNA extensions being shorter than the 30nt required for DIS3-

mediated processing via the exosome core[66,69]. Our present experimental set up is not optimised to identify which of the suggested scenarios is most likely to take place, but new CLIP technology promises to be able to identify NEXT assembly- and disassembly events, possibly revealing more detail[27].

## Methods

### Cell culture
HeLa Kyoto cell lines containing stable integrations of LAP-tagged ALYREF, CBP80 or RBM7 were used for tiCLIP experiments and were sourced from Poser et al.[40] The LAP tag contains a GFP and S peptide separated by a TEV cleavage site. BAC integrations of the whole gene locus resulted in expression of the tagged protein at near endogenous levels[40]. Standard HeLa cells were used for ALYREF immunoprecipitation analyses and immunofluorescence experiments. Both types of HeLa cells were cultured in Dulbecco's modified eagle medium (DMEM; Invitrogen), supplemented with 10% foetal bovine serum and 1% penicillin/streptavidin, and incubated at 37 °C and with 5% $CO_2$, unless stated otherwise.

### DRB-mediated transcription synchronisation
Cells were grown to 70% confluency. To initiate the DRB block, the existing media was replaced with preconditioned media containing DRB (100µM) and incubated for 3 hrs. To release the DRB block, the DRB-containing media was aspirated and cells were washed twice with PBS before adding fresh media. Next, the cells were returned to the incubator for the appropriate time frame and then UV cross-linked. For timepoint 0, samples were UV cross-linked (see below) without replacing the media. For the non-synchronised control (DMSO), cells were incubated with DMSO-containing media for 3 hrs prior to UV cross-linking.

### UV cross-linking
Ice-cold PBS was used for washes and cell collection. 20 sec before the end of the timepoint, the cell media was rapidly aspirated and followed by two washes with PBS to remove residual media. Next, the minimum volume of PBS required to cover the cells was added to the cell culture plate. The cell culture plate was then placed on a bed of ice and UV cross-linked for 30 sec using 254 nm (UVC) irradiation (Stratagene). UV cross-linked samples were lifted from plates using cell scrapers before collecting in PBS. These samples were then pelleted, snap frozen and stored at −80 °C.

### CLIP library construction
The steps for tiCLIP library construction were inspired by[70]. Key oligonucleotides and reagents can be found in Supplementary Table 1, and recipes for buffers referenced within this section can be found in Supplementary Table 2. Cell pellets were resuspended in 2 mL of lysis buffer (LB) supplemented with Ribolock (2.5 µL/mL), protease Inhibitors (1×), SDS (0.1%) and DTT (1 µM). On ice, cells were lysed by sonication. Protein concentrations were normalised and made up to 2 mL in volume. Next, cell lysates were treated with DNaseI [5 µL/mL; ThermoFischer] and RNase I [50 U/mL; Ambion] for exactly 5 min at 37 °C and 1200 rpm. Immediately upon completion samples were stored on ice for 5 min. Lysates were cleared by centrifugation at 16,000 × g for 20 min at 4 °C. Next, 100 µL of GFP-Trap (Chromotek) bead slurry was prewashed with LB and added to 1.9 mL of cleared lysate and rotated for 1 hr at 4 °C at 16 rpm. Bead slurries were then washed three times with LB, two times with high salt wash (HSW), two times with LiCl Buffer (LCW), one time with no-salt buffer (NSB), and one time with dephosphorylation buffer (DPB). Next, on bead dephosphorylation of RNAs from recovered ribonucleoproteins was performed using FastAP (ThermoFisher), supplemented with Ribolock (ThermoFisher) for 20 min at 37 °C at 1200 rpm. The dephosphorylation mix was aspirated while bead slurries were magnetically precipitated and then washed

twice with phosphate wash buffer (PWB) and two times with 1× ligation buffer (LigBx1). Next, on-bead ligation was performed using pre-adenylated barcoded L3. Beads were incubated with 2 µL of 10× ligation buffer (Tris-HCl 500 mM; $MgCl_2$ 100 mM), 5 µL PEG-400, 0.5 µL RNase Inhibitor (Promega), 1 µL T4 RNA ligase 2 truncated KQ (NEB) and 2 µL of the appropriate pre-adenylated barcoded L3 linker (10 µM) and incubated for 16 hrs at 16 °C with shaking at 1250 rpm for 15 sec every 1.5 min. Beads were precipitated on a magnet and linker ligation mixes were aspirated. Beads were washed one time with HSW, two times with LB and one time with polynucleotide kinase wash (PNKW). RNAs recovered from RNPs were radiolabelled with gamma-ATP using polynucleotide kinase (NEB) for 45 mins at 37 °C. Phosphorylation was terminated by adding 100 µL of phosphatase buffer mix and radioactive PNK mix was aspirated whilst beads were precipitated on magnets. Radiolabelled RNPs were washed three times with NSW and resuspended in 20 µL of 1.5× NuPAGE loading buffer and incubated for 10 mins at 70 °C using 1100 rpm. Following this, 1 µL DTT (1 M) was added, and microfuge tubes were incubated for a further 5 mins at 95 °C with no shaking. The radiolabelled RNPs were then separated using SDS-PAGE and transferred to a nitrocellulose membrane and detected by exposing to x-ray films. Autoradiograms were used to guide isolation of radiolabelled RNPs from the nitrocellulose membrane. Sections of membranes were cut at least 15 kDa above the expected migration of the un-crosslinked protein as recovery of RNPs above this range ensured that RNA:L3 barcoded adapters were at least 50nt in length[71], which was compatible with sequencing and downstream analysis. Excised membrane pieces were incubated with 400 µL of PK-buffer and 10 µL of proteinase K (Fischer Scientific) for 2 hrs at 55 °C to release cross-linked RNAs. RNAs were then extracted using phenol-chloroform and precipitated using EtOH and sodium acetate (pH 5.5). cDNAs were synthesised using Superscript IV reverse transcriptase (ThermoFisher) using a RtCLIP primer barcoded primer. Thereafter, RNA was removed from reverse transcriptase reactions by alkaline hydrolysis, and cDNA was precipitated using EtOH and sodium acetate. Next, cDNAs were separated on a 1× TBE Urea 6% polyacrylamide gel (Novex) and 3 bands were excised from the gel: high (135nt to 215nt), middle (100nt to 135nt) and low (85 nt to 100 nt), which represent cDNA lengths of 75nt to 150nt, 35nt to 75nt, 35nt to 20nt, respectively. Gel pieces were placed in a 1.5 mL microfuge tube and crushed using a 1 mL syringe plunger. 400 µL of TE buffer was added to the crushed gel pieces and the sample was snap frozen before incubating at 37 °C for 2 hrs with 1200 rpm shaking. Gel pieces were removed from the eluate by centrifugation through columns (Costar) plugged with 2 glass filters. The eluate was precipitated using EtOH and sodium acetate. Next, the PCR template was prepared by circularising the cDNA using CircLigse II (Epicentre) for 1 hr at 60 °C. This intra-molecular ligation reaction ligated the 5'adapter contained in the RtCLIP reverse transcriptase primer to the 3'end of the cDNA. Next, cDNA circles were incubated with 1 µL of cut oligo [10 µM] by heating to 95 °C for 1 min, and then decreasing the temperature every 20 sec by 1 °C until 25 °C was reached. Next, 1 µL of BamHI (Fast Fermentas) was added and samples were incubated for 30 min at 37 °C, followed by 5 min 80 °C. Linearised cDNAs were precipitated using EtOH and sodium acetate. Finally, PCR libraries were generated by using Solexa_P5 and Solexa_P3 primers required for Illumina sequencing and amplified by Accuprime Supermix enzyme (Invitrogen). High, middle and low cDNA samples were precipitated using AmpureXP beads to remove primers and mixed 5:5:1 ratio. All tiCLIP libraries were pooled before sequencing using 75 bp paired-end sequencing.

### On-bead RNase I digestion
For RNase titration assays displayed in Supplementary Fig. 1d, all cell lysates from samples were subjected to RNase I treatment for 5 min (in-lysate), followed by a post-immunoprecipitation RNase treatment which was performed 'on-bead'. To perform the additional 'on-bead'

RNaseI digestion, the above CLIP library construction protocol was completed up to the dephosphorylation of RNA 'on-bead' step. Here, the beads from one immunoprecipitation were equally split across 5 microfuge tubes and resuspended in 200 μL of LB (lysis buffer) supplemented with the appropriate concentration of RNaseI. RNase digestions were incubated at 37 °C with 1200 rpm agitation for 3 min and halted by adding 800 μL of HS wash and placing on ice immediately. Cross-linked RNAs were processed and analysed as described in the CLIP library construction protocol above, but omitting the L3 linker ligation.

## Barcoded pre-adenylated L3 DNA adapters
Previously tested in-line barcodes (Supplementary Data 1 and Supplementary Table 1) were placed at the 5′ end of the original iCLIP L3 DNA adapter[71,72]. Oligos were 5′ adenylated using 5′ DNA adenylation kit (NEB), purified using ssDNA/RNA Clean & Concentrator columns (Zymo Research), and diluted to 10 μM before storing at −20 °C till required.

## Western blotting analysis
Equal recovery of GFP-tagged proteins and, when appropriate, their co-precipitants was confirmed by western blotting analysis. After isolating protein-RNA complexes, the membrane was incubated with 5% skimmed milk powder (SMP) in PBST (0.05% Tween) for 1 hr and incubated with primary antibodies (Supplementary Table 1) diluted in 5% SMP in PBST. The following dilutions for primary antibodies were used: anti-GFP 1:1000; anti-CBP80 1:1000; anti-EIF4A3 1:1000; anti-MLN51 1:1000, anti-Y14 1:500; anti-MAGOH 1:500. Subsequently, membranes were washed three times for 20 min in PBST and incubated for 30 mins with horseradish peroxidase (HRP) conjugated antibodies diluted to 1:10,000 with 5% SMP in PBST. Membranes were washed again and exposed using Supersignal West Femto Substrate (ThermoFischer).

## Immunofluorescence analysis
Cells were grown to 70% of confluence and fixed with 4% paraformaldehyde in PBS for 20 min at room temperature, washed 2× in PBS and stored in 70% EtOH at 4 °C. Cells were rehydrated in PBS and permeabilized in 0.5% triton X-100 in PBS for 10 min at room temperature. A blocking step was performed with 2% BSA in PBS for 30 min at room temperature. The primary antibody (mouse monoclonal anti-GFP (B-2); Santa Cruz)was diluted 1:500 in PBS and incubated for 1 h at room temperature, washed three times with PBS, and incubated with secondary antibody (Alexa 488 conjugated Goat Anti-mouse IgG (H + L); ThermoFisher), which was diluted 1:1000 in PBS. Cells were imaged on a Zeiss AxioObserver microscope with a Hxp120V light source, an Axiocam 702 mono camera and a Plan-Neofluar 40 × 0.75 NA objective.

## Library demultiplexing, mapping and quality control
Raw reads were demultiplexed and processed using a combination of the pyCRAC software package tool pyBarcodeFilter.py[73] and TrimGalore (https://doi.org/10.5281/zenodo.5127899). The tiCLIP sample information and their associated L3 adapter and RtCLIP primer barcode sequences can be found in Supplementary Data 1. Full adapter sequences can be found in Supplementary Table 1. Adaptor trimmed reads passing both length and quality cut-offs were then mapped to GRCh38 (Ensembl) using Hisat2[74], using default parameters with the following options: --rna-strandness FR −fr −phred33 −no-softclip. Unmapped and non-primary reads were discarded using SAMtools[75]. Mapped PCR duplicates were removed using UMI barcodes and the first mapping coordinate[76]. After assessing the raw mapped reads, CBP20-3 was omitted from the analyses due to a consistently low number of mapped reads across all timepoint and samples (Supplementary Data 1).

## CBP20 and CBP80 read length analysis
Approximately 70 nt of RNA will cause cross-linked RBP migration to be retarded by 20 kDa[71]. Thus, a minimum RNA length, in cross-linked CBP20-LAP RNPs, required to retard it enough to co-migrate with isolated CBP80 RNPs would be 228nt (or 199nt when taking into account the 29nt adapter sequence ligated prior to running SDS-PAGE). This calculation was made given that the intervening distance between CBP80 and CBP20-LAP migrations was 65 kDa (Supplementary Fig. 1e; compare lower boundary of upper red box vs. single arrow in middle panel). To explore RNA lengths, we analysed the relevant tiCLIP libraries and extracted the observed template length (TLEN) from bam files and found that the median insert size for CBP20 vs. CBP80 libraries differed by only 1–8nt, which was insufficient to explain a bulk of longer RNA fragments cross-linked to CBP20 populating the CBP80 libraries.

## Cross-link and 3′CLIP read position extrapolation
Protein-RNA cross-link sites were extracted from mapped reads by shifting the 5′position of read1 1nt upstream. The last nucleotide of read2 was extracted and flipped to the opposite strand in order to extrapolate 3′CLIP signal. Read1 was used for whole read plots. Manipulations were processed using a combination of bedtools[77].

## Library and expression normalisation, and count file generation
To account for the differing amounts of RNA-protein interactions recovered from tiCLIP timepoints, we used rRNA read counts to normalise the libraries sizes, rather than read per million (RPM), which assumes all libraries have equal RNA inputs. For each tiCLIP library, the counts of reads mapping to 20 rRNA annotations (ENSG00000199839, ENSG00000202264, ENSG00000199523, ENSG00000199480, ENSG00000199415, ENSG00000201059, ENSG00000278189, ENSG00000200558, ENSG00000201321, ENSG00000199994, ENSG00000200408, ENSG00000272435, ENSG00000274917, ENSG00000272351, ENSG00000201185, ENSG00000210082, ENSG00000211459, ENSG00000275215, ENSG00000275757, ENSG00000276700) were summed and divided by 30,000 (arbitrary number) to create an rRNA factor (Supplementary Data 1), which was then used to scale bedGraph files, that were used for subsequent normalised coverage plots and normalised counts. bedGraph files for each timepoint and strand were created using bedtools genomecov[77] and the rRNA factor was used as an input for the -scale option. To calculate tiCLIP coverage over TUs or specific regions of the genome at nucleotide resolution, awk scripts, bedtools intersect, and bedtools map were used to map read counts to bed files harbouring the regions of interest or TUs.

## TU annotation and gene expression normalisation
TUs annotations were derived from a flattened HeLa cells transcriptome[46]. For data presented in tiCLIP coverage plots, iCLIP coverage was normalised to gene expression from a previous RNAseq experiment conducted in HeLa cells[46].

## Spatiotemporal RNA-binding heatmaps
TUs were stratified into groups based on their length; specifically, at increments of 10 kb, starting at 0 to 10 kb and ending at 290 kb to 300 kb. TUs above 300 kb were grouped. Next, individual TUs were segmented into 1 kb chunks progressing from the TSSs to the TESs. Reads mapping to TU chunks were counted using bedtools[77] and the mean number of reads mapping to each bin was normalised to the max value of the TU group length.

## k-means clustering analysis and coverage profiles for mature RNAs
CLIP coverage profiles for mature TUs (mature RNAs) were generated by calculating the relative distance from the TSSs when only

considering exonic segments. TUs were normalised for length by dividing each mature TU into 100 equally sized bins. Cross-link positions within each bin were aggregated and divided by the sum of all bins for that given TU. Only TUs >200nt in length with >20 mapped CLIP reads were considered. Displayed is the average profile of all TUs. K-means clustering analysis was performed on all of the TU profiles using 3 centres for each biological replicate of ALYREF-DMSO independently. The results from each biological replicate were intersected to produce bona fide TU profile groups.

### RNA 3'seq pA+/pA− data analysis
RNA 3'-end-seq pA+/pA− data generated from HeLa cell total RNA after depletion of the exosome core component RRP40 or treated with control siRNA were downloaded from GEO Series accession number GSE137612[18]. The specific bigwig files were sourced from the following accession numbers GSM4083151, GSM4083150, GSM4083149, GSM4083132, GSM4083131 and GSM4083130. Briefly, bigwig files were converted to bedGraph files and then intersected with snoRNA annotation bed files, containing coordinates for the snoRNA 100nt up- and downstream regions, using bedtools intersect and map subcommands[77]. The resultant count files were converted to coverage profiles using custom R code employing tidyr and ggplot2 packages.

### Statistical analysis
Where statistical tests were used, their identity is indicated in the appropriate figure legend. Values were considered significant when $p < 0.05$.

### Reporting summary
Further information on research design is available in the Nature Portfolio Reporting Summary linked to this article.

## Data availability
The datasets generated during this study have been deposited in NCBI's Gene Expression Omnibus (GEO)[78] and are accessible through GEO Series accession number GSE202980. The minimum datasets that are necessary to interpret, verify and extend our research have been deposited Zenodo (https://doi.org/10.5281/zenodo.7535697). Relevant processed data are included as Source Data. The RNA 3'seq data was published by Wu et al.[18] and is accessible under GEO series accession number GSE137612. The ALYREF CLIP performed in conjunction with the knockdown of CBP80 or PABPN1 was published in Shi et al., 2017[23] and is accessible under GEO series accession number GSE99069. Source data are provided in this paper.

## Code availability
The computational analyses were conducted using R or bash scripting, which has been deposited at https://doi.org/10.5281/zenodo.7535646.

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

## Acknowledgements

We thank Dr. Will Garland and Dr. Søren Lykke-Andersen for critically reading the manuscript. Work in the T.H.J. laboratory was funded by the Lundbeck and Novo Nordisk foundations (NNF, ExoAdapt Grant 31199). R.A.C. was supported by an EMBO long-term fellowship (ALTF 1070-2017) and a Marie Curie Individual Fellowship (797358). S.G. was supported by a Medical Research Council Non-Clinical Senior Research Fellowship (MR/R008205/1). We thank Dr. Ina Poser for the generous gifts of the HeLa cell lines expressing LAP-tagged proteins and Dr. Maria Gockert for RNA 3′seq analysis guidance.

## Author contributions

R.A.C., S.G., and T.H.J. designed the experiments. R.A.C., Y.D., R.T., and A.B. performed the experiments. R.A.C. performed bioinformatic analyses. Y.D. performed western analyses of ALYREF-LAP precipitates and A.B. performed western analyses of LAP-tagged and their counterpart endogenous proteins. R.T. performed immunofluorescence. T.H.J., supervised the project. R.A.C. and T.H.J. wrote the manuscript with input from all co-authors.

## Competing interests

The authors declare no competing interests.
