## [Peer Review File · Nature Communications]

Temporal iCLIP captures co-transcriptional RNA-protein interactionsREVIEWER COMMENTS

Reviewer #1 (Remarks to the Author):

Temporal iCLIP captures co-transcriptional RNA-protein interactions

by Cordiner et al.

In their manuscript, Cordiner and colleagues develop a new approach to study RBP binding in a time-resolved and genome-wide manner. To achieve this, they inhibit transcription elongation with DRB which stalls polymerases at the transcription start sites. Upon removal of DRB, transcription resumes in a coordinated way. Protein binding to the newly emerging RNAs is then measured with iCLIP in a time course experiment. Hence, RBP binding can be studied at transcripts that have been synthesized since DRB removal for various timepoints. The authors use this approach to study four proteins that are part or associated with the cap-binding complex: CPB20, CBP80, RBM7 and ALYREF. The authors find that initial, possibly unspecific binding occurs immediately upon transcription, while more specific binding patterns arise at later timepoints.

I think this is overall a very interesting approach and it is exciting to learn how RBPs associate with nascent RNA in cells. However, with the current study I have three major concerns that will be detailed below: 1) I am not sure if quality of the datasets is sufficient to make certain conclusions. 2) A large part of the data is driven by the emergence of the nascent RNA, however this has not been analysed or controlled for. 3) None of the observations has been tested with orthogonal approaches or functional assays. I think publication of this manuscript would require significant improvement on these issues.

Major comments:

1) I appreciate that for this study, a large number of datasets have been generated. However, I am not sure if these data are of sufficient quality to draw conclusions about spurious vs. specific binding. As I understand from the table, most libraries have less than 1 million reads per sample. This is very little for quantitative comparisons. In particular for investigations with nascent RNA that have to cover a 10 times bigger sequence space than for mature RNA. To identify binding sites and to differentiate them from spurious binding, some form of peak calling would be required. I am not sure if this is possible with the current datasets.

Also, currently it is difficult to judge the quality of the data by reading the manuscript. It will be important for the reader to give more numbers throughout the text. For example, how many mapped reads are available per sample after PCR duplicate removal? How many genes are usually analysed in the different categories (indicate in figures)? The table at the end is difficult to read, but this could be due to the formatting.

Also, I suggest that the authors show some genome browser views depicting raw data, similar to the schematic shown in Figure 1b. It will be important to show that the global observations also hold for individual examples and are not only due to aggregation.

2) As the authors conclude themselves, a lot of the RBP iCLIP signal reflects unspecific binding to the nascent transcriptome appearing upon DRB removal. A major complication is that this nascent transcriptome changes for each timepoint. Hence, it is unclear if changes in iCLIP signal reflect changes in RBP binding or changes in the transcriptome available for binding. In order to draw reliable conclusions, it will be essential to monitor changes in the nascent transcriptome. Comparison of changes in iCLIP signal to changes in the nascent transcriptome will then inform on changes in RBP binding.

3) All analyses are based on the tiCLIP which of course provides a big source of data. However, this is a new form of data, and it is not clear to what extent the conclusions made are reliable. Hence, orthogonal validation or some form of functional assay to validate predictions would be required.

Minor comments:

According to a previous publication (<https://doi.org/10.1038/sj.emboj.7600876>), transcription of short intron-less histone and snRNA-encoding genes is not affected by CDK9 inhibition. It would be important to confirm that the genes referred to as intron-less in the current study are indeed inhibited by DRB.

Figure 1c, d: It is not explained what "negative" means.

I think that the abbreviation "IP" has not been introduced and I would refrain from using the verb "IP'ed".

Reviewer #2 (Remarks to the Author):

In this manuscript, the authors describe an in-depth analysis of time resolved series of iCLIP experiments, an approach that they call tiCLIP. The chosen query RNA-binding factors are three proteins known as early nuclear RNA binding factors: ALYREF, an RNA export adaptor, RBM7, component of the NEXT complex involved in targeting the nuclear exosome to its substrates and CBC20 and CBC80, factors of the Cap Binding Complex.

While the contribution of including a time dimension to the analysis is not always obvious, in several instances it turned out to provide some determinant clues to the chronology of the RNP maturation processes.

Importantly, while the detailed analyses of the experiments remain complex and often not straightforward to follow, it turns out that, somehow paradoxically, adding the time dimension rather simplifies the interpretation of the results.

Altogether, the description of this novel implementation to the CLIP approach, the sophisticated and in-depth analyses of this complex body of data and the resulting conclusions that can be drawn for it contribute to make this paper interesting and provides a nice and important addition to the CLIP approach.

There are two points, which I think could improve the manuscript if correctly addressed:

- Lines 305 to 310. It seems to me that this statement would be much clearer if the authors would also provide the 3'CLIP meta-analysis displayed in fig.3h for spliced exons, across the junction. Indeed, one could wonder if the sudden fall of the signal in the intron region does not simply reflect the fact that ALYREF preferentially binds to spliced exons (first and internal exons, fig. 3g), unlike CBC. I thus think that it is important to make sure that there is not a continuum in the 3'CLIP signal for ALYREF when aligned on spliced exons.

- Fig. 6c, bottom: the discrete nature of the pics, in particular the 3 pics downstream of the H/ACA 3'ends, is intriguing. While the proposed explanation is very plausible, one would like to be sure that this is not the result of a frequent bias in this type of analyses, i.e. that the averaged coverage could be dominated by a single or very few, highly abundant snoRNAs. Could some single snoRNA examples be provided in the supplemental data. Alternatively, the averaged coverage could be calculated after a normalization of snoRNA derived sequences in such a way that each snoRNA would equally contribute to the aggregated data.

(very) minor point:

Fig. 5d: I presume that the orange cartouche in the scheme represents the point of crosslink? This should be specified in the figure legend.

Reviewer #3 (Remarks to the Author):

Temporal iCLIP captures co-transcriptional protein-RNA interactions

by Cordiner et al.

Cordiner et al. investigate the time-resolved binding of four central nuclear RNA-binding proteins to nascent transcripts. For this, the authors developed a new protocol, tiCLIP, in which iCLIP is combined with release from transcriptional inhibition. They observe a progressive binding along the transcripts, illustrating how RBPs sample RNAs co-transcriptionally to find their steady-state binding sites. The authors use these data to investigate the mechanisms of RBP deposition, such as the splicing-dependent deposition of ALYREF, RBM7 binding to intron lariats, and RBM7 association during snoRNA processing.

Investigating the dynamics of RBP binding during co-transcriptional RNP assembly is important to understand gene expression regulation. The presented protocol and data allow for a detailed description of these processes. The authors perform a series of elaborate analyses on their data, which lead to interesting observations. However, no orthogonal data are provided to test any of the proposed mechanisms, and the study thereby stays rather descriptive at this point (see below).

The manuscript is very well written, with detailed explanations of all steps, and nicely presented figures.

Major point:

1. The conclusions in the manuscript are based exclusively on the tiCLIP data for the four RBPs. At present, none of the conclusions are tested further. The authors should consider to include orthogonal data to support some of the proposed mechanistic links. This would considerably strengthen the manuscript.

2. The authors use GFP-tagged proteins for the tiCLIP experiments. From the included citation, it appears that these are BAC-mediated overexpression (?) cell lines. I could not find information in the Methods on how these cell lines were generated or validated.

Since the conclusions in this manuscript strongly rely on the tiCLIP experiment, the authors need to validate the correct expression of the GFP-tagged constructs. For instance, they should use microscopy to confirm the correct subcellular localisation of the tagged proteins and test their expression levels relative to the endogenous proteins using Western blots. In addition, the authors should demonstrate, if possible, that overexpression of the GFP-tagged variants does not impair protein functionality, e.g., by testing for differences in known target genes.

3. The authors obtained an "indirect profile" for CBP80 from the CBP20 tiCLIP experiments, because both proteins were pulled down together in the conditions used. I wonder how the presence of the two proteins in the IP may affect the specificity of the obtained libraries. E.g., would the lower band smear into the upper band, such that CBP20-RNA complexes with longer RNA fragments appear in the CBP80 library? This should be discussed.

In this context, it is also unclear how the regions cut from the membrane were chosen in Supplementary Figure 1a. Why was such a slim band chosen for CBP20, which lies much above the labelled expected height of the protein? Conversely, in the case of RBM7, the signal extends well below the labelled expected height of the protein, which was apparently taken into account when cutting from way below the size of the protein. Where does this signal come from?

The authors need to describe these aspects in more detail and perform additional controls where possible. For instance, to justify the chosen regions, the authors should test high RNase conditions for all proteins to show the condensed bands of the proteins crosslinked to RNA.

Minor points:

1. It would be helpful to provide exemplary genome browser views to visualise data quality and to

support the validity of the conclusions.

The data have apparently been submitted to GEO (<https://www.ncbi.nlm.nih.gov/geo/query/acc.cgi?&acc=GSE202980>) but I did not find a secure token to access the submission.

2. The authors normalise the tiCLIP libraries to the included rRNA read counts to account for overall changes in RNA binding. This is an interesting approach which may be applicable in many other studies. The underlying assumption is that the rRNA content is generally stable and does not underlie random fluctuations between replicates. Also, are there sufficient numbers of rRNA reads detected to allow for reliable estimates? As a quality control, the authors should show the relative amount of rRNA reads in each sample. Also, the changes in the normalisation factors over time could be visualised.

3. "CBP20's binding in close proximity to RNA 5'caps prevented its RNA-mediated 5'phosphate labelling". Is there a reference that could be cited here or is this a conclusion from the present analyses?

4. Figure 4c: In the panel "Exons", the y-axis is labelled as "log2 size (nt)", but text and legend imply that the plot shows the absolute number of exons.

5. Figure 6a,b: Why were whole reads taken here rather than crosslink sites?

Typo:

Line 792: Circlagse

Dear reviewers,

We thank you for the insightful and considerate reviews of our manuscript. Below we provide a point-by-point response to detail how the manuscript has been improved during its revision. As a general comment, we note that Fig. 3d-f and Fig. 5a-b have been slightly modified due to a minor bug in the original code used for producing these figures. The fix gave rise to slightly different y-axis numbers, but overall, the figures are unchanged. Furthermore, we have also corrected some wording, typos and added additional descriptions in the methods where necessary are highlighted in cyan. Changes to the manuscript and figure legends, that were in response to reviewers' comments, are highlighted in yellow. Additionally, we have included source data of uncropped western blots and immunofluorescence images as part of this submission. Finally, we have provided a link to code used for analysis; this can be found in the 'Data and code availability' section present in the manuscript.

All the best,

Ross and Torben

Reviewer #1 (Remarks to the Author):

Temporal iCLIP captures co-transcriptional RNA-protein interactions

by Cordiner et al.

In their manuscript, Cordiner and colleagues develop a new approach to study RBP binding in a time-resolved and genome-wide manner. To achieve this, they inhibit transcription elongation with DRB which stalls polymerases at the transcription start sites. Upon removal of DRB, transcription resumes in a coordinated way. Protein binding to the newly emerging RNAs is then measured with iCLIP in a time course experiment. Hence, RBP binding can be studied at transcripts that have been synthesized since DRB removal for various timepoints. The authors use this approach to study four proteins that are part or associated with the cap-binding complex: CPB20, CBP80, RBM7 and ALYREF. The authors find that initial, possibly unspecific binding occurs immediately upon transcription, while more specific binding patterns arise at later timepoints.

I think this is overall a very interesting approach and it is exciting to learn how RBPs associate with nascent RNA in cells. However, with the current study I have three major concerns that will be detailed below:

- 1) I am not sure if quality of the datasets is sufficient to make certain conclusions.
- 2) A large part of the data is driven by the emergence of the nascent RNA, however this has not been analysed or controlled for.
- 3) None of the observations has been tested with orthogonal approaches or functional assays. I think publication of this manuscript would require significant improvement on these issues.

A/ We appreciate the reviewer's comments and provide a detailed response to the specific points below.

Major comments:

1) I appreciate that for this study, a large number of datasets have been generated. However, I am not sure if these data are of sufficient quality to draw conclusions about spurious vs. specific binding. As I understand from the table, most libraries have less than 1 million reads per sample. This is very little for quantitative comparisons. In particular for investigations with nascent RNA that have to cover a 10 times bigger sequence space than for mature RNA. To identify binding sites and to differentiate them from spurious binding, some form of peak calling would be required. I am not sure if this is possible with the current datasets.

A/ Excluding the negative samples, we have generated tiCLIP libraries from 86 protein and timepoint combinations. We agree with the reviewer that the coverage achieved is not always deep enough for quantitative comparisons of discrete binding sites within individual transcripts, however, our approach was also not intended to do this. Instead, we aimed to expose general spatiotemporal binding profiles of the tested RBPs from a meta-analysis perspective of protein-RNA interactions across all genes at each timepoint and stratified by gene size (Fig. 2a). This produced a qualitative wave of RNA binding, that invaded into the gene body at a rate comparable with RNAPII transcription (a wave which could also be captured on individual transcripts (new Fig. S2a-b)). With this starting point, we aimed to identify generally specific binding profiles over common transcript landmarks. We believe that our approach

has achieved this original intention with the sufficient data depth and temporal resolution (Figs 3,5,6).

Our experiments were not set up to identify signal peaks, but straightforward parallel analyses of the tested protein's CLIP density profiles readily identified non-overlapping patterns. For example, RBM7 binds downstream of snoRNA 3'ends and upstream of intron 3'ends, CBP20/80 binds at RNA 5'ends and ALYREF shows increased binding 25nt upstream of the 3'ends of exons. All of these observations are compatible with previously published CLIP dataset (Giacometti et al., 2017; Lubas et al., 2015; Shi et al., 2017; Tuck and Tollervey, 2013; Viphakone et al., 2019). We agree with the reviewer, that more direct conclusions drawn about nascent RNA require larger datasets. However, we note that the tiCLIP approach should not be compared to nascent RNAseq. tiCLIP leverages RNAPII dynamics (e.g. pausing, followed by its release) to capture the spatiotemporal binding of RBPs across successively increasing windows of nascent transcription. This contrasts nascent RNAseq, which leverages high sequencing depth in combination with saturating the transcriptome with metabolic labelling, usually 4sU, to assess the nascent transcriptome at steady-state. That being said, we are happy to 'tone-down' our conclusions about initial spurious binding if the reviewer prefers.

Also, currently it is difficult to judge the quality of the data by reading the manuscript. It will be important for the reader to give more numbers throughout the text. For example, how many mapped reads are available per sample after PCR duplicate removal? How many genes are usually analysed in the different categories (indicate in figures)? The table at the end is difficult to read, but this could be due to the formatting.

A/ We agree that this information was somewhat sparse and gene numbers have now been embedded in all relevant figure or added to their legends. In addition, we have added additional columns in Supplementary Table 1 to aid interpretation. We now present the number of mapped reads after the quality control steps of collapsing PCR duplicates and removing secondary alignments. Moreover, we show counts of mapped reads and the percentages of initial mapped reads.

Also, I suggest that the authors show some genome browser views depicting raw data, similar to the schematic shown in Figure 1b. It will be important to show that the global observations also hold for individual examples and are not only due to aggregation.

A/ We have now added genome browser views to the following figures:

- 1) Supplementary Figure 2a-b – depicting regions with mapped reads. Referenced at line 224.*
- 2) Supplementary Figure 4b-c – depicting TUs from 'group1' or 'group2' as identified from our k-means clustering analysis of ALYREF binding profiles. Referenced within the main text at line 365 and 366.*
- 3) Supplementary Figure 5a-b – depicting lariat binding hallmarks revealed by RBM7 cross-link and 3'CLIP profiles. Referenced within the main text at line 439.*

Finally, we have included bar plots, showing the individual nucleotide coverage of RBM7 tiCLIP data over the 3'region of 4 snoRNAs in (new Supplementary Fig. 6a) and is referenced at line 507.

2) As the authors conclude themselves, a lot of the RBP iCLIP signal reflects unspecific

binding to the nascent transcriptome appearing upon DRB removal. A major complication is that this nascent transcriptome changes for each timepoint. Hence, it is unclear if changes in iCLIP signal reflect changes in RBP binding or changes in the transcriptome available for binding. In order to draw reliable conclusions, it will be essential to monitor changes in the nascent transcriptome. Comparison of changes in iCLIP signal to changes in the nascent transcriptome will then inform on changes in RBP binding.

A/ In order to capture RBP binding during transcription, we paused transcription for 3 hours, before its re-initiation. In doing so, we have effectively emptied the cell of its nascent RNA content by allowing processing of all but the longest primary transcripts to go to completion. These conditions are of course artificial and some RBPs will be mis-localized due to the shutdown of transcription. However, while keeping this premise in mind, our experiments do uncover RBP-RNA binding profiles, that progressively build up over the time course to the expected, and published, steady-state binding patterns. For this reason, we believe that the new and more detailed RNA binding profiles identified by our metagene analysis are likely to be relevant. We see this, for example, for the RBM7 tiCLIP data, that uncover a reverse transcription truncation pattern, leading us to conclude that RBM7 can bind to lariats, prior to the second transesterification step of splicing (Fig 5). Similar iCLIP signals have been reported for SF3 in Briese et al., 2019. We also identify RBM7 binding downstream of snoRNA 3'ends, and at the 3'ends of introns (Fig. 6 and as previously identified in Lubas et al., 2015). For these areas where we have depicted nucleotide resolution coverage; the tiCLIP binding profiles do suggest the binding is specific to a discrete region, and is not transient association due to the transcription of nascent RNA.

The reviewer suggests a nascent RNA control, that would aid in normalising the tiCLIP data. This would be extremely complicated to generate. To capture nascent RNA in a similar way, the experiment in question would require:

- 1) DRB-block for 3h, followed by metabolic labelling with 4sU.*
- 2) RNA harvested at 0, 5, 10, 15, 20, 40 and 60 min time points.*
- 3) Complete purification of labelled RNA away from total RNA.*
- 4) Sequencing of both 'labelled' and 'total' RNA samples.*

Although purification of labelled RNA from total RNA is a convenient way to isolate newly synthesised RNA, it also presents a number of drawbacks, such as contamination with up to 30% of total RNA present in labelled fractions (Rabani et al., 2014) and diminishing returns of recovered RNA, common especially for the short labelling pulses required to match our early time points (0-20 min). Moreover, integration of total and nascent RNA-seq data requires their own normalisations.

For the above reasons we decided that a simpler option would be to normalise each tiCLIP library to its rRNA content (see Methods). We used this method as standard reads per million (RPM) normalisation, that is generally applied to CLIP or RNA-seq libraries, cannot be used here as the transcriptomes for each of the timepoints are not at steady-state.

3) All analyses are based on the tiCLIP which of course provides a big source of data. However, this is a new form of data, and it is not clear to what extent the conclusions made are reliable. Hence, orthogonal validation or some form of functional assay to validate predictions would be required.

A/ We have included additional and orthogonal data in the revised manuscript as follows:

i) To corroborate our finding that ALYREF can be recruited to nascent RNA before the second transesterification step of splicing, we have now included a western blotting analysis of ALYREF-LAP immunoprecipitates (new Fig. S3e). We find that the protein co-precipitates three core members of the exon-junction complex (EJC,) whose recruitment to RNA was previously shown to coincide with the first transesterification step of splicing (Gehring et al., 2009). This supports the notion that ALYREF is recruited to the splicing reaction at the suggested time. Additionally, we have incorporated the following sentence into the main text at line number 622:

“Gratifyingly, the first transesterification step of splicing also coincides with the recruitment of the exon junction complex (EJC) components MAGHO, Y14 and eIF4A3 (Gehring et al., 2009) (Supplementary Fig. 3e)”

ii) To corroborate our suggestion that the RBM7 3'CLIP data demarcate the limits of RBM7-mediated 3'-5' end processing of H/ACA- and CD-box snoRNAs by the RNA exosome, we consulted previously published 3'end RNAseq data captured from HeLa cells depleted of the core exosome component RRP40. Indeed, increased abundances are observed of RNA 3'ends 9nt and 25nt downstream of H/ACA- and CD-box snoRNAs, respectively. In contrast, such increases are absent upstream of the snoRNAs. This lends clear support to the notion that the RNA exosome is required for processing of intron-residing snoRNAs up to these discrete sites. We have joined this analysis as a new Supplementary Fig. 6b and added the following text to the main text section at line number 515:

“In support of this notion, re-analysis of RNA 3'end-seq data, capturing both polyadenylated (pA⁺) and unadenylated (pA⁻) transcripts from HeLa cells depleted of the core exosome component RRP40 (Wu et al., 2020a, and see Methods), revealed an increased abundance of 3'ends immediately downstream of the 9nt and 25nt peaks identified by RBM7 3'CLIP analysis of H/ACA- and CD-box snoRNAs, respectively (Supplementary Fig. 6b). As similar increases were not present upstream of these peaks, we suggest they represent blocks to further exosome/RBM7-mediated processing of H/ACA- and CD-box snoRNAs (Fig. 6g).”

Additionally, we have added an extra section to the methods, and added the relevant information to the “Data and code availability” section.

Minor comments:

According to a previous publication (<https://doi.org/10.1038/sj.emboj.7600876>), transcription of short intron-less histone and snRNA-encoding genes is not affected by CDK9 inhibition. It would be important to confirm that the genes referred to as intron-less in the current study are indeed inhibited by DRB.

A/ The reviewer is correct that DRB does not affect transcription of the short snRNA- and intron-less histone-genes. We have therefore repeated the analysis relating to Fig. 1d-e and Supplementary Fig 1f-h after filtering out snRNA and histone genes. Although this did not result in any qualitative changes to the produced graphs or quantitative differences between time points, we have, for completeness, now included these new analyses, and have added the following text to the manuscript at line 170:

“We excluded snRNA- and replication dependent histone (RDH)-TUs mRNAs from this analysis as their transcription is unaffected by CDK9 inhibitors, such as DRB (Medlin et al., 2005).”

Concerning the issue of whether mono-exonic genes might generally be unaffected by DRB: of the four tiCLIP profiles generated, the only protein whose binding was present after the DRB block was RBM7. However, this was independent of exon content and RBM7 tiCLIP read density on mono-exonic genes increased 2.5-fold from 4 to 10 reads/kb (Fig. 1d; RBM7 and mono-exonic panel). A similar increase was also observed for multi-exonic genes. This temporal increase demonstrates that DRB is indeed blocking transcription of mono- and multi-exonic transcripts.

Figure 1c, d: It is not explained what “negative” means.

A/ Thank you. We have now added:

“‘Negative’ timepoints represent controls in which blank magnetic beads were used (negative anti-GFP lanes) on unsynchronised samples.”

I think that the abbreviation “IP” has not been introduced and I would refrain from using the verb “IP’ed”.

A/ We have replaced two instances of “IP’ed” with “immunoprecipitated”.

Reviewer #2 (Remarks to the Author):

In this manuscript, the authors describe an in-depth analysis of time resolved series of iCLIP experiments, an approach that they call tiCLIP. The chosen query RNA-binding factors are three proteins known as early nuclear RNA binding factors: ALYREF, an RNA export adaptor, RBM7, component of the NEXT complex involved in targeting the nuclear exosome to its substrates and CBC20 and CBC80, factors of the Cap Binding Complex. While the contribution of including a time dimension to the analysis is not always obvious, in several instances it turned out to provide some determinant clues to the chronology of the RNP maturation processes.

Importantly, while the detailed analyses of the experiments remain complex and often not straightforward to follow, it turns out that, somehow paradoxically, adding the time dimension rather simplifies the interpretation of the results.

Altogether, the description of this novel implementation to the CLIP approach, the sophisticated and in-depth analyses of this complex body of data and the resulting conclusions that can be drawn for it contribute to make this paper interesting and provides a nice and important addition to the CLIP approach.

There are two points, which I think could improve the manuscript if correctly addressed:

1) Lines 305 to 310. It seems to me that this statement would be much clearer if the authors would also provide the 3'CLIP meta-analysis displayed in fig.3h for spliced exons, across the junction. Indeed, one could wonder if the sudden fall of the signal in the intron region does not simply reflect the fact that ALYREF preferentially binds to spliced exons (first and internal exons, fig. 3g), unlike CBC. I thus think that it is important to make sure that there is not a continuum in the 3'CLIP signal for ALYREF when aligned on spliced exons.

A/ This is an excellent suggestion. An additional panel is now provided in Supplementary Fig. 3, where we have stitched together 1st and 2nd exons to display the 3'CLIP signals for CBP20, CBP80 and ALYREF across both exons. For visual purposes, we have also separated the timepoints. As can now be appreciated, there is no continuous exon 1 signal bleeding into exon 2 (Supplemental Figure 3d). This is now explained by the newly added sentence at line number 330:

“However, it did not form a continuum of signal into the downstream exon, which would be expected of a protein preferentially binding to spliced RNA (Supplementary Fig. 3d). This demonstrated that ALYREF can be recruited before the second transesterification step of splicing,”

2) Fig. 6c, bottom: the discrete nature of the pics, in particular the 3 pics downstream of the H/ACA 3'ends, is intriguing. While the proposed explanation is very plausible, one would like to be sure that this is not the result of a frequent bias in this type of analyses, i.e. that the averaged coverage could be dominated by a single or very few, highly abundant snoRNAs. Could some single snoRNA examples be provided in the supplemental data. Alternatively, the averaged coverage could be calculated after a normalization of snoRNA derived sequences in such a way that each snoRNA would equally contribute to the aggregated data.

A/ Following the suggestion from the reviewer, we have now added a heatmap of the 3'CLIP data across a 35nt window directly downstream of the snoRNA 3'ends. Here, we show the top 50 H/ACA- and CD-box snoRNAs, as judged by the sum of all 3'CLIP reads mapping to the 35nt window, to demonstrate that the observations are generalised across a large number of individual snoRNAs (new Figure 6e-f). Additionally, we have included histogram coverage plots of 4 representative snoRNAs

in Supplementary Fig. 6a. These changes are accompanied by the following text addition at line number 505:

“These RBM7-derived 3’CLIP profiles were representative of multiple individual CD- and H/ACA-box snoRNAs (Fig. 6e-f and Supplementary Fig. 6a).”

Adjust figure

(very) minor point:

Fig. 5d: I presume that the orange cartouche in the scheme represents the point of crosslink? This should be specified in the figure legend.

A/ We have added the label “Proteinase K digestion” to Figure 5d, and also clarified this in the legend of Figure 5d:

“Cross-linked protein is represented by orange cartouche, whereas the orange triangle represents a short peptide cross-linked to the RNA that remains after Proteinase K treatment.”

Reviewer #3 (Remarks to the Author):

Temporal iCLIP captures co-transcriptional protein-RNA interactions
by Cordiner et al.

Cordiner et al. investigate the time-resolved binding of four central nuclear RNA-binding proteins to nascent transcripts. For this, the authors developed a new protocol, tiCLIP, in which iCLIP is combined with release from transcriptional inhibition. They observe a progressive binding along the transcripts, illustrating how RBPs sample RNAs co-transcriptionally to find their steady-state binding sites. The authors use these data to investigate the mechanisms of RBP deposition, such as the splicing-dependent deposition of ALYREF, RBM7 binding to intron lariats, and RBM7 association during snoRNA processing.

Investigating the dynamics of RBP binding during co-transcriptional RNP assembly is important to understand gene expression regulation. The presented protocol and data allow for a detailed description of these processes. The authors perform a series of elaborate analyses on their data, which lead to interesting observations. However, no orthogonal data are provided to test any of the proposed mechanisms, and the study thereby stays rather descriptive at this point (see below).

The manuscript is very well written, with detailed explanations of all steps, and nicely presented figures.

Major point:

1. The conclusions in the manuscript are based exclusively on the tiCLIP data for the four RBPs. At present, none of the conclusions are tested further. The authors should consider to include orthogonal data to support some of the proposed mechanistic links. This would considerably strengthen the manuscript.

A/ We have included additional and orthogonal data in the revised manuscript as follows:

i) To corroborate our finding that ALYREF can be recruited to nascent RNA before the second transesterification step of splicing, we have now included a western blotting analysis of ALYREF-LAP immunoprecipitates (new Fig. S3e). We find that the protein co-precipitates three core members of the exon-junction complex (EJC,) whose recruitment to RNA was previously shown to coincide with the first transesterification step of splicing (Gehring et al., 2009). This supports the notion that ALYREF is recruited to the splicing reaction at the suggested time. Additionally, we have incorporated the following sentence into the main text at line number 622:

“Gratifyingly, the first transesterification step of splicing also coincides with the recruitment of the exon junction complex (EJC) components MAGHO, Y14 and eIF4A3 (Gehring et al., 2009) (Supplementary Fig. 3e)”

ii) To corroborate our suggestion that the RBM7 3'CLIP data demarcate the limits of RBM7-mediated 3'-5' end processing of H/ACA- and CD-box snoRNAs by the RNA exosome, we consulted previously published 3'end RNAseq data captured from HeLa cells depleted of the core exosome component RRP40. Indeed, increased abundances are observed of RNA 3'ends 9nt and 25nt downstream of H/ACA- and CD-box snoRNAs, respectively. In contrast, such increases are absent upstream of the snoRNAs. This lends clear support to the notion that the RNA exosome is required for

processing of intron-residing snoRNAs up to these discrete sites. We have joined this analysis as a new Supplementary Fig. 6b and added the following text to the main text section at line number 515:

“In support of this notion, re-analysis of RNA 3’end-seq data, capturing both polyadenylated (pA⁺) and unadenylated (pA⁻) transcripts from HeLa cells depleted of the core exosome component RRP40 (Wu et al., 2020a, and see Methods), revealed an increased abundance of 3’ends immediately downstream of the 9nt and 25nt peaks identified by RBM7 3’CLIP analysis of H/ACA- and CD-box snoRNAs, respectively (Supplementary Fig. 6b). As similar increases were not present upstream of these peaks, we suggest they represent blocks to further exosome/RBM7-mediated processing of H/ACA- and CD-box snoRNAs (Fig. 6g).”

Additionally, we have added an extra section to the methods, and added the relevant information to the “Data and code availability” section.

2. The authors use GFP-tagged proteins for the tiCLIP experiments. From the included citation, it appears that these are BAC-mediated overexpression (?) cell lines. I could not find information in the Methods on how these cell lines were generated or validated.

Since the conclusions in this manuscript strongly rely on the tiCLIP experiment, the authors need to validate the correct expression of the GFP-tagged constructs. For instance, they should use microscopy to confirm the correct subcellular localisation of the tagged proteins and test their expression levels relative to the endogenous proteins using Western blots. In addition, the authors should demonstrate, if possible, that overexpression of the GFP-tagged variants does not impair protein functionality, e.g., by testing for differences in known target genes.

A/ Good point. We have now included expression level (western blotting) and subcellular localisation (immunofluorescence) analyses of the LAP-tagged proteins (Supplementary Fig. 1a-b). Gratifyingly, we find that all fusion proteins are nuclear localised. ALYREF-LAP is under expressed when compared to endogenous ALYREF, whereas RBM7-LAP and CBP20-LAP are mildly overexpressed when compared to their endogenous counterparts. The RBM7-LAP and CBP20-LAP expression cell lines interaction partners have been previously characterised by mass spectrometry analysis and were (Andersen et al., 2013; Dou et al., 2020).

Additionally, we have included western blotting analyses of an ALYREF-LAP IP experiment (new Supplementary Fig. 3e), showing that the fusion protein exhibits RNA-independent interactions with core components of the EJC, which is in-line with previous studies (Gromadzka et al., 2016; Viphakone et al., 2019).

3. The authors obtained an “indirect profile” for CBP80 from the CBP20 tiCLIP experiments, because both proteins were pulled down together in the conditions used. I wonder how the presence of the two proteins in the IP may affect the specificity of the obtained libraries. E.g., would the lower band smear into the upper band, such that CBP20-RNA complexes with longer RNA fragments appear in the CBP80 library? This should be discussed.

A/ This is indeed a valid point. Our estimate that 70nt of RNA equals app. 20kDa was ascertained from Huppertz et al., 2014, who isolated cross-linked RNA from 20kDa above the RBP of interest and confirmed that the minimum RNA size was 70nt. Thus, the minimum length of cross-linked RNA required to retard the migration of GFP-CBP20, for it to co-migrate with CBP80:RNA complexes, could be approximated: the lower cut for isolating CBP80-RNA was at 115kDa (Supplementary Fig. 1e; upper red box in middle panel), which is 65kDa above the expected migration of the CBP20-LAP

protein of 50kDa. Therefore the minimum expected RNA size, which would potentially co-migrate with CBP20:RNA complexes is 228nt (70nt/20kDa x 65kDa) or 199nt of RNA when taking into account the 29nt adapter sequence ligated prior to running SDS-PAGE. To explore RNA lengths cross-linked to cap-binding proteins further, we analysed the relevant *ti*CLIP libraries and found that the median insert size for CBP20 vs. CBP80 libraries differed by only 1-8nt. Hence, this is insufficient to explain a bulk of longer RNA fragments cross-linked to CBP20 populating the CBP80 libraries. We suggest this minimal change towards larger insert sizes might be due to the larger range cut from the membrane for CBP80:RNA (CBP80: 115kDa – 185kDa vs CBP20: 60kDa-75kDa).

We have now incorporated this analysis as part of Supplementary Fig. 1, in new panel f, and updated the figure legend. Additionally we have added the following text to the manuscript at line number 158:

“To ensure that the indirect *ti*CLIP profile for CBP80 was not deriving from cross-linked CBP20 RNP, we explored RNA lengths cross-linked to these proteins. The median insert size for CBP20 vs. CBP80 libraries differed by only 1-8nt (Supplementary Fig. 1f), which was insufficient to retard CBP20 RNPs and cause co-migration with CBP80 (see Methods).”

And the following text to the methods at line number 842:

“CBP20 and CBP80 read length analysis

Approximately 70nt of RNA will cause cross-linked RBP migration to be retarded by 20kDa (Huppertz et al., 2014). Thus, a minimum RNA length, in cross-linked CBP20-LAP RNPs, required to retard it enough to co-migrate with isolated CBP80 RNPs would be 228nt (or 199nt when taking into account the 29nt adapter sequence ligated prior to running SDS-PAGE). This calculation was made given that the intervening distance between CBP80 and CBP20-LAP migrations was 65kDa (Supplementary Fig. 1e; compare lower boundary of upper red box vs. single arrow in middle panel). To explore RNA lengths, we analysed the relevant *ti*CLIP libraries and extracted the observed template length (TLEN) from bam files and found that the median insert size for CBP20 vs. CBP80 libraries differed by only 1-8nt, which was insufficient to explain a bulk of longer RNA fragments cross-linked to CBP20 populating the CBP80 libraries.”

In this context, it is also unclear how the regions cut from the membrane were chosen in Supplementary Figure 1a. Why was such a slim band chosen for CBP20, which lies much above the labelled expected height of the protein? Conversely, in the case of RBM7, the signal extends well below the labelled expected height of the protein, which was apparently taken into account when cutting from way below the size of the protein. Where does this signal come from?

A/ The expected migrations of the proteins shown in Supplementary Fig 1d-e are as follows:

- GFP-CBP20 ~50kDa
- CBP80 ~80kDa
- GFP-ALYREF ~65kDa
- GFP-RBM7 ~65kDa.

The solid arrows, which marked the expected protein migrations, had unfortunately been shifted when making the figure. We apologise for this oversight and have now

fixed it. Also, for reference please note the western blots below the autoradiograms with protein size markers shown.

A slim “CBP20-RNA band“ was cut to recover crosslinked RNA long enough for sequencing and mapping. Given that a ribonucleotide on average accounts for 0.5kDa, an RNA of 30nt cross-linked to the protein of interest retards its migration by 15kDa. A calculation was made to cut the blot at least 15kDa above the expected molecular weight. Following this, and in order to not encroach into protein:RNA complexes recovered from CBP80, a thin band between 65kDa – 70kDa was cut for GFP-CBP20. This has now been explained in the Supplementary Fig. 1 legend and a section has been added to the Methods at line number 759:

“Sections of membranes were cut at least 15kDa above the expected migration of the un-crosslinked protein as recovery of RNPs above this range ensured that RNA:L3 barcoded adapters were at least 50nt in length (Huppertz et al., 2014), which was compatible with sequencing and downstream analysis.”

The authors need to describe these aspects in more detail and perform additional controls where possible. For instance, to justify the chosen regions, the authors should test high RNase conditions for all proteins to show the condensed bands of the proteins crosslinked to RNA.

A/ RNaseI conditions were optimised prior to running the full experiment. In sum, we found that 50U/mL of in-lysate digestion produced the strongest autoradiogram signals. We also performed high RNaseI titrations for all of the LAP-immunoprecipitates. To do this, an initial in-lysate RNaseI digestion was followed by a second RNaseI digestion. This was performed with LAP-immunoprecipitates still bound to the anti-GFP beads and after stringent washing, which proved to be a more extensive RNaseI digestion than its ‘in-lysate’ counterpart. Indeed, this showed a shift from a diffuse radiolabelled smear above the expected molecular weight towards a more concentrated band of the expected migration. Additionally, we silver stained precipitates from the LAP-tag protein IPs, which demonstrated robust IP conditions.

All of these data are now included in Supplementary Fig. 1c-d and have been referenced from the main text with the following at line number 147:

“Near-endogenous expression, correct subcellular localisation, and robust immunoprecipitation conditions for all of the engaged LAP-tagged proteins were confirmed (Supplementary Fig. 1a-d). “

Minor points:

1. It would be helpful to provide exemplary genome browser views to visualise data quality and to support the validity of the conclusions.

A/ We have now added genome browser views to the following figures:

- 1) Supplementary Figure 2a-b – depicting regions with mapped reads. Referenced at line 224.*
- 2) Supplementary Figure 4b-c – depicting TUs from ‘group1’ or ‘group2’ as identified from our k-means clustering analysis of ALYREF binding profiles. Referenced within the main text at line 365 and 366.*

3) *Supplementary Figure 5a-b – depicting lariat binding hallmarks revealed by RBM7 cross-link and 3'CLIP profiles. Referenced within the main text at line 439.*

Finally, we have included bar plots, showing the individual nucleotide coverage of RBM7 tiCLIP data over the 3' region of 4 snoRNAs in (new Supplementary Fig. 6a) and is referenced at line 507.

The data have apparently been submitted to GEO (<https://www.ncbi.nlm.nih.gov/geo/query/acc.cgi?&acc=GSE202980>) but I did not find a secure token to access the submission.

A/ We have generated an access token `efwbsyaextyjbqh`.

2. The authors normalise the tiCLIP libraries to the included rRNA read counts to account for overall changes in RNA binding. This is an interesting approach which may be applicable in many other studies. The underlying assumption is that the rRNA content is generally stable and does not underlie random fluctuations between replicates. Also, are there sufficient numbers of rRNA reads detected to allow for reliable estimates? As a quality control, the authors should show the relative amount of rRNA reads in each sample. Also, the changes in the normalisation factors over time could be visualised.

A/ Indeed, we normalised to rRNA read counts due to its high levels (~95% of stable cellular RNA originates from rDNA loci). We chose the 30 rRNA annotations that had the most mapped reads across all tiCLIP samples with the assumption that the in-parallel conducted IPs would contain approximately the same amounts of background rRNA. The normalisation factors for each replicate sample are shown in Supplementary Table 1, 'rRNA Factor' column. Additionally, we have now included the exact numbers of rRNA reads for each sample and the respective rRNA content as a % of the whole library. Ranges of rRNA content were between 0.1-3%, with the highest rRNA content stemming from the negative IPs. We have now provided a graph (see below) within this rebuttal to summarise the distribution of rRNA content among all timepoints, stratified by immunoprecipitated protein.

Figure 1 – % of mapped reads that map to rRNA. Stratified by Protein.

3. “CBP20’s binding in close proximity to RNA 5’caps prevented its RNA-mediated 5’phosphate labelling”. Is there a reference that could be cited here or is this a conclusion from the present analyses?

A/ The weak radioactive signal above the GFP-CBP20 band suggested that, although CBP20 binds to RNA, it may not be visualised by the tiCLIP protocol, using 5’ radiolabelling of copurified RNA substrates. We drew our conclusion from the biochemistry of the enzymes used in the tiCLIP protocol and previous biochemical studies that show CBP20-RNA interactions (Calero et al., 2002; Mazza et al., 2002). CBP20 interacts directly with the m7Gppp cap. As RNase digestion is performed in-lysate, before denaturing the RNP complexes, the CBC-RNP is most likely shielded and co-purified RNA may consequently contain the 5’capped part of the transcript bound to CBP20. The alkaline phosphatase enzyme used does not convert trimethyl caps into 5’OHs, which are the substrates for 32P radiolabelling. We have added text to Supplementary Fig. 1d to rationalize this:

“The lack of radioactive signal associated with the CBP20-RNPs (middle panel, single black arrow) is likely explained by cross-linking of CBP20 close to the 5’ cap, preventing radiolabelling as the alkaline phosphatase enzyme does not convert trimethyl caps into the necessary 5’OH substrates. “

4. Figure 4c: In the panel “Exons”, the y-axis is labelled as “log2 size (nt)”, but text and legend imply that the plot shows the absolute number of exons.

A/ We thank the reviewer for noting this mistake. The correct y-axis is log2 exon number, which has now been corrected.

5. Figure 6a,b: Why were whole reads taken here rather than crosslink sites?

A/ When analysing processed RNA, we considered it more appropriate to include whole read data as these depict the RBM7-bound processing intermediates better. For reference, the crosslink data is present in Supplementary Fig. 6d-e.

Typo:
Line 792: CircLagse

A/ Thanks for spotting this typo, which has now been corrected.

Bibliography

- Andersen, P.R., Domanski, M., Kristiansen, M.S., Storvall, H., Ntini, E., Verheggen, C., Schein, A., Bunkenborg, J., Poser, I., Hallais, M., et al. (2013). The human cap-binding complex is functionally connected to the nuclear RNA exosome. *Nat. Struct. Mol. Biol.* *20*, 1367–1376.
- Briese, M., Haberman, N., Sibley, C.R., Faraway, R., Elser, A.S., Chakrabarti, A.M., Wang, Z., König, J., Perera, D., Wickramasinghe, V.O., et al. (2019). A systems view of spliceosomal assembly and branchpoints with iCLIP. *Nat. Struct. Mol. Biol.* *26*, 930–940.
- Calero, G., Wilson, K.F., Ly, T., Rios-Steiner, J.L., Clardy, J.C., and Cerione, R.A. (2002). Structural basis of m7GpppG binding to the nuclear cap-binding protein complex. *Nat. Struct. Biol.* *9*, 912–917.
- Dou, Y., Barbosa, I., Jiang, H., Iasillo, C., Molloy, K.R., Schulze, W.M., Cusack, S., Schmid, M., Le Hir, H., Lacava, J., et al. (2020). NCBP3 positively impacts mRNA biogenesis. *Nucleic Acids Res.* *48*, 10413–10427.
- Gehring, N.H., Lamprinak, S., Hentze, M.W., and Kulozik, A.E. (2009). The hierarchy of exon-junction complex assembly by the spliceosome explains key features of mammalian nonsense-mediated mRNA decay. *PLoS Biol.* *7*.
- Giacometti, S., Benbahouche, N.E.H., Domanski, M., Robert, M.-C., Meola, N., Lubas, M., Bunkenborg, J., Andersen, J.S., Schulze, W.M., Verheggen, C., et al. (2017). Mutually Exclusive CBC-Containing Complexes Contribute to RNA Fate. *Cell Rep.* *18*, 2635–2650.
- Gromadzka, A.M., Steckelberg, A.L., Singh, K.K., Hofmann, K., and Gehring, N.H. (2016). A short conserved motif in ALYREF directs cap- and EJC-dependent assembly of export complexes on spliced mRNAs. *Nucleic Acids Res.* *44*, 2348–2361.
- Huppertz, I., Attig, J., D’Ambrogio, A., Easton, L.E., Sibley, C.R., Sugimoto, Y., Tajnik, M., König, J., and Ule, J. (2014). iCLIP: Protein-RNA interactions at nucleotide resolution. *Methods* *65*, 274–287.
- Lubas, M., Andersen, P.R., Schein, A., Dziembowski, A., Kudla, G., and Jensen, T.H. (2015). The Human Nuclear Exosome Targeting Complex Is Loaded onto Newly Synthesized RNA to Direct Early Ribonucleolysis. *Cell Rep.* *10*, 178–192.
- Mazza, C., Segref, A., Mattaj, I.W., and Cusack, S. (2002). Large-scale induced fit recognition of an m7GpppG cap analogue by the human nuclear cap-binding complex. *EMBO J.* *21*, 5548–5557.
- Medlin, J., Scurry, A., Taylor, A., Zhang, F., Peterlin, B.M., and Murphy, S. (2005). P-TEFb is not an essential elongation factor for the intronless human U2 snRNA and histone H2b genes. *EMBO J.* *24*, 4154–4165.
- Rabani, M., Raychowdhury, R., Jovanovic, M., Rooney, M., Stumpo, D.J., Pauli, A., Hacohen, N., Schier, A.F., Blackshear, P.J., Friedman, N., et al. (2014). High-resolution sequencing and modeling identifies distinct dynamic RNA regulatory strategies. *Cell* *159*, 1698–1710.
- Shi, M., Zhang, H., Wu, X., He, Z., Wang, L., Yin, S., Tian, B., Li, G., and Cheng, H. (2017). ALYREF mainly binds to the 5′ and the 3′ regions of the mRNA in vivo. *Nucleic Acids Res.* *45*, 9640–9653.
- Tuck, A.C., and Tollervey, D. (2013). A transcriptome-wide atlas of RNP composition reveals diverse classes of mRNAs and lncRNAs. *Cell* *154*, 996–1009.
- Viphakone, N., Sudbery, I., Griffith, L., Heath, C.G., Sims, D., and Wilson, S.A. (2019). Co-transcriptional Loading of RNA Export Factors Shapes the Human Transcriptome. *Mol. Cell* *75*, 310-323.e8.

Wu, G., Schmid, M., Rib, L., Polak, P., Meola, N., Sandelin, A., and Jensen, T.H. (2020). A Two-Layered Targeting Mechanism Underlies Nuclear RNA Sorting by the Human Exosome. *Cell Rep.* 30, 2387-2401.e5.

REVIEWERS' COMMENTS

Reviewer #1 (Remarks to the Author):

I would like to thank the authors for a very clear author response to the comments. All my concerns were fully addressed. Congratulations on a very nice study.

Reviewer #2 (Remarks to the Author):

The authors have, in my opinion, very satisfactorily replied to the referee's comments and modified the manuscript accordingly.

Reviewer #3 (Remarks to the Author):

Temporal iCLIP captures co-transcriptional protein-RNA interactions
by Cordiner et al.

The authors have addressed all comments and answered all questions satisfactorily. I congratulate the authors to an interesting study and suggest the manuscript for publication in Nature Communication.